


# Transport of Po Valley aerosol pollution to the northwestern Alps. Part 1: phenomenology

Henri Diémoz[1], Francesca Barnaba[2], Tiziana Magri[1], Giordano Pession[1], Davide Dionisi[2],
Sara Pittavino[1], Ivan K. F. Tombolato[1], Monica Campanelli[2], Lara Sofia Della Ceca[3], Maxime Hervo[4],
Luca Di Liberto[2], Luca Ferrero[5], and Gian Paolo Gobbi[2]

[1]ARPA Valle d'Aosta, Saint-Christophe, Italy
[2]Institute of Atmospheric Science and Climate, CNR, Rome, Italy
[3]Instituto de Física Rosario, Rosario, Argentina
[4]MeteoSwiss, Payerne, Switzerland
[5]University of Milan-Bicocca, Milan, Italy

*Correspondence to:* Henri Diémoz (h.diemoz@arpa.vda.it)

**Abstract.**

Mountainous regions are often considered pristine environments, however they can be affected by pollutants emitted in more populated and industrialised areas, transported by regional winds. Based on experimental evidence, further supported by modelling tools, we demonstrate and quantify here the impact of air masses transported from the Po Valley, a European atmo-

spheric pollution hotspot, to the northwestern Alps. This is achieved through a detailed investigation of the phenomenology of near-range (few hundreds km), trans-regional transport, exploiting synergies of multi-sensor observations mainly focussed on particulate matter. The explored dataset includes vertically-resolved data from atmospheric profiling techniques (Automated LiDAR-Ceilometers, ALC), vertically-integrated aerosol properties from ground (sun photometer) and space, and in situ measurements ($PM_{10}$ and $PM_{2.5}$, relevant chemical analyses, and aerosol size distribution). During the frequent advection

episodes from the Po basin, all the physical quantities observed by the instrumental setup are found to significantly increase: the scattering ratio from ALC reaches values >30, AOD triplicates, surface $PM_{10}$ reaches concentrations $> 100\,\mu g\,m^{-3}$ even in rural areas, secondary inorganic compounds such as nitrate, ammonium and sulfate increase up to 28%, 8% and 17% of the total $PM_{10}$ mass, respectively. Results also indicate that the advected aerosol is smaller in size and less light-absorbing compared to the aerosol type locally-emitted in the northwestern Italian Alps, and hygroscopic. In this work, the phenomenon

is exemplified through detailed analysis and discussion of three case studies, selected for their clarity and relevance within the wider dataset, the latter being fully exploited in a companion paper quantifying the impact of this phenomenology over the long- term (Diémoz et al., 2018). For the three case studies investigated, a high-resolution numerical weather prediction model (COSMO) and a lagrangian tool (LAGRANTO) are employed to understand the meteorological mechanisms favouring the transport and to demonstrate the Po Valley origin of the air masses. In addition, a chemical transport model (FARM) is

used to further support the observations and to partition the contributions of local and non-local sources. Results show that the simulations are not able to adequately reproduce the measurements (with modelled $PM_{10}$ concentrations 4–5 times lower than the ones retrieved from the ALC, and maxima anticipated by 6–7 hours), likely owing to deficiencies in the emission inventory



and particle water uptake not fully taken into account. The advected aerosol is shown to remarkably degrade the air quality of the Alpine region, with potential negative effects on human health, climate and ecosystems, as well as on the touristic development of the investigated area. The findings of the present study could also help design mitigation strategies at the trans–regional scale in the Po basin, and suggest an observations-based approach to evaluate the outcome of their implementation.

## 1 Introduction

In mountainous regions, mutual exchanges between the valley atmosphere and the nearby plains have been recognised and studied for more than a century (e.g., Thyer, 1966, and references therein). Notably, daytime up-valley (nighttime down-valley) flows systematically develop as a result of faster heating (cooling) of mountain valleys compared to the foreland (Rampanelli et al., 2004; Serafin and Zardi, 2010; Schmidli, 2013; Wagner et al., 2014), and hence manifest on a very regular basis, especially during fair-weather days (nights) with weak synoptic circulation (Borghi and Giuliacci, 1980; Tampieri et al., 1981). The plain-to-mountain circulation regime conveys mass, heat and moisture within the planetary boundary layer (PBL), thus contributing to horizontal mixing on the mesoscale (Weissmann et al., 2005). Additionally, air parcels can be lifted by convection above the ridges and transported to the free troposphere, which favours air mass exchange in the vertical direction (Henne et al., 2004; Gohm et al., 2009; Schnitzhofer et al., 2009; Lang et al., 2015).

Thermally-driven wind systems are observed in mountainous regions throughout the world (e.g., Cong et al., 2015; Collaud Coen et al., 2018; Dhungel et al., 2018). The European Alps have been the ideal scenario for such kind of studies, owing to their rugged shape forming hundreds of main and tributary valleys, and large surrounding plains with strong emission sources, the most significant being in the Po basin. Indeed, this vast region, which includes a large portion of northern Italy, is one of the most densely populated (more than 20 millions of people and a population density of 414 inhabitants per $km^2$ (WMO, 2012)), industrialised, and thus polluted areas in Europe (Chu et al., 2003; Van Donkelaar et al., 2010; Fuzzi et al., 2015; EMEP, 2016). The valley morphology exacerbates the air quality. In fact, heavy emissions from productive activities as well as from vehicular traffic and residential heating are often trapped within the Po basin due to its characteristic topography strongly limiting the dispersion of pollutants, with the Alpine chain and the Apennines enclosing the plain on its northern, western and southern sides. As a consequence, the Po basin is one of the EU hotspots suffering from premature mortality associated to atmospheric pollution (EEA, 2015). In spite of the improvements in the last decades (e.g., Bigi and Ghermandi, 2016), the air quality in the Po Valley is still far from the standards established by the European Commission (EU Commission, 2008) and exceedances of these standards are expected to continue in the next years (Belis et al., 2017; Caserini et al., 2017; EEA, 2017; Guariso and Volta, 2017).

Both theoretical studies and experimental campaigns demonstrated that transboundary transport of several kind of pollutants from the Po basin affects pre-Alpine areas (Dosio et al., 2002; Neftel et al., 2002; Mélin and Zibordi, 2005), the Italian Alpine valleys (Nyeki et al., 2002; Larsen et al., 2012; Ferrero et al., 2014), other Italian regions (Cristofanelli et al., 2009; Carbone et al., 2014; Moroni et al., 2015) and even neighbouring countries (e.g., Wotawa et al., 2000; Finardi et al., 2014). Over the impacted areas, a correct partitioning between local and non-local sources is therefore necessary to 1) correctly interpret the





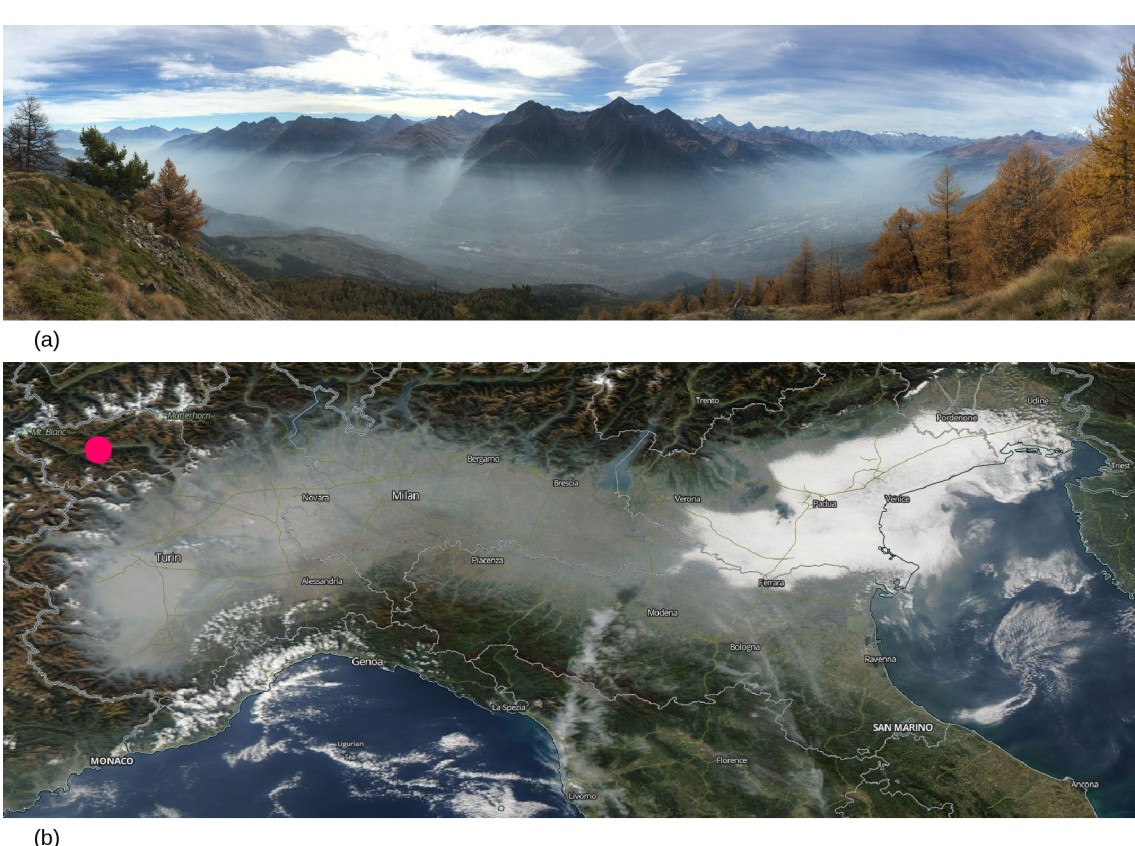

**Figure 1.** (a) Panoramic view of the main valley over the Aosta–Saint Christophe station during a pollution advection event. The picture was taken from Croce di Fana (2200 m a.s.l, Quart village, 6 km north-east of Aosta–Saint Christophe) on 21 October 2017. On that day, the advected layer of aerosol – visible in the picture as a hazy layer – reached an altitude of about 2000 m a.s.l. Photo kindly provided by C. Cometto. (b) Image of the Po Valley from the MODIS Aqua radiometer (corrected reflectance, true colour) only few days before the picture in the first panel was taken (18 October, https://worldview.earthdata.nasa.gov). The satellite view clearly shows that the hazy, aerosol-rich layer from the Po basin is starting to pour out into the Alpine valleys. The pink marker identifies the Aosta–Saint Christophe site.




exceedances of air quality limits; 2) develop joint efforts and large-scale mitigation strategies (WMO, 2012) to reduce the frequency and impact of pollution episodes (an example is the recently signed "antismog" agreement for the improvement of air quality in the Po basin, Italian Ministry of the Environment (2018)); 3) assess the impact of air pollutants on citizen health (Straif et al., 2013; WHO, 2016; Zhang et al., 2017), climate (Clerici and Mélin, 2008; Lau et al., 2010; Zeng et al., 2015)

and ecosystems (Carslaw et al., 2010; Bourgeois et al., 2018; Burkhardt et al., 2018). As an additional important aspect, in mountainous regions, pollution layers undermine the visual quality of the landscape (e.g., Fig. 1a) and touristic attractiveness, with obvious economics implications (de Freitas, 2003).

In the present study, we aim at illustrating and deeply investigating the phenomenology of aerosol transport events to the northwestern Alpine region through a detailed analysis and discussion of specifically-selected case studies. The impacts of

this phenomenon over the long-term are then quantified in a companion paper (Diémoz et al., 2018). This research exploits a multi-technique approach, combining a large set of measurements (at surface level, column integrated and vertically-resolved) with modelling tools such as chemical transport models (CTM). In fact, several previous field campaigns investigated the atmospheric composition and transport mechanisms in the eastern and central part of the Po Valley (Nyeki et al., 2002; Barnaba et al., 2007; Ferrero et al., 2010; Larsen et al., 2012; Decesari et al., 2014; Khan et al., 2016; Rosati et al., 2016; Cugerone

et al., 2018), but very few studies are available on the westernmost side of the basin (Anfossi et al., 1988; Mercalli et al., 2003; Manara et al., 2018). Earlier evidences of possible advections of pollutants from the Po basin to the northwestern Alps were collected in the framework of two intensive, 4-days-long campaigns performed between 2000 and 2001 (Agnesod et al., 2003). At that time, an equipped aircraft flew during anticyclonic conditions with weak synoptic circulation, in order to assess the effects of the local winds on the air quality in the Alpine valleys close to Mont Blanc (in Italy, France and Switzerland).

The experiment was focussed on ozone measurements and its precursors, however aerosol concentrations were additionally measured. Capping inversions limiting the development of the mixing layer, vertical transport of pollutants along the valley slopes, and the ozone-polluted residual layer aloft (entrained into the mixing layer during the next day) were the most interesting phenomena explored by that study. Advections from the Po Valley with thermally-driven flows were hypothesised to be the main factor contributing to the high ozone concentrations found in the elevated layers. Although this result was supported by

measurements of carbon monoxide, ambient particulate matter (PM) and relative humidity, the short duration of the campaign could not allow to exclude other effects. More recently, Diémoz et al. (2014a) analysed a one-year-long time series of columnar aerosol optical properties measured by a sun/sky photometer in the same area and found that the heaviest burden of particles did not come from urban settlements within the valley, but rather from outside the valley, namely the Po basin.

Overall, this work attempts to answer the following scientific questions still lacking a comprehensive understanding:

1. What is the origin of the aerosol layers detected in the northwestern Alps?

2. What conditions are favourable to the aerosol flow into the valley?

3. How do the advected aerosol layers evolve in both altitude and time?

4. What is the impact of the transported aerosol on PM surface concentrations and chemical composition?





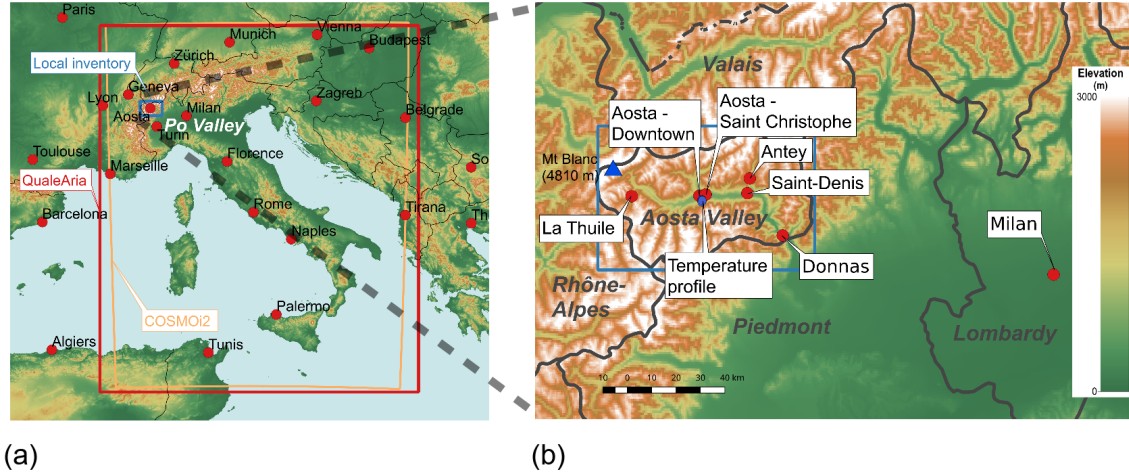

(a)                (b)

**Figure 2.** (a) Elevation map of Italy, showing the position of the investigated Alpine region and the geographical domains of: 1) the Numerical Weather Prediction model (COSMO-I2, orange box), 2) the national emission inventory (QualeAria, red box) and 3) the local inventory of the Aosta Valley (blue box). (b) Zoom over the Aosta Valley and northwestern Italy, with location of the measurement stations. The blue box corresponds to the geographic domain of the local inventory as in panel (a). The elevation (colour) scale is common to both figures.

5. Are the current chemical transport models able to reproduce and explain the observations at the ground and along the vertical profile?

    Though referred to the location object of the study, these questions are of more general interest, as several regions of the world are characterized by basin valleys surrounded by mountains. Hence, the role of pollution advections, their vertical behaviour and the final impact at ground-level is a matter of global interest.

    The paper is organised as follows: the investigated area is presented in Sect. 2, while Sect. 3 describes both the experimental (3.1) and the modelling (3.2) approach used. Results (Sect. 4) are presented by addressing specific case studies to exemplify the advection of polluted, aerosol-rich air masses and comparing them to the simulations. Conclusions are drawn in Sect. 5.

## 2   Investigated area and experimental sites

This study is mainly focussed on the Aosta Valley, the smallest Italian administrative region (130000 inhabitants, Fig. 2). It is about 80 km by 40 km wide, and is located in the northwestern side of the Alps, not far from the two major urban settlements and industrial areas of the Po Valley, i.e. Turin (80 km) and Milan (150 km). The region is characterised by a complex topography, typical of the Alpine valleys. Its surface elevation varies from 300 to 4800 m a.s.l. (average altitude > 2000 m a.s.l.), with several tributary valleys starting from the main valley. The latter connects Mt. Blanc (at the border with France, Fig. 2) to the Piedmont region through a 90 km-long directrix, approximately divisible into three segments with NW-SE, W-E and NW-SE directions. This main valley is narrower at both ends (with minimum width of few hundreds meters) and





widens in correspondence of the Aosta city, the largest urban settlement of the region (about 35000 inhabitants). The complex topography triggers several meteorological phenomena typical of mountain valleys, such as wind channelling along the main valley, thermally-driven winds from the plain to the mountains (and vice versa), rain-shadow (foehn) winds and temperature inversions. The latter are very frequent during wintertime, occurring about 50% of the time (Vuillermoz et al., 2013). Not

surprisingly, these dynamics strongly affect the dispersion of pollutants and the air quality in the lowest atmospheric layers.

Data from several measuring sites in the Aosta Valley are used in this work (Fig. 2b). Most of the instruments employed (Sect. 3.1) are operated at the two observatories run by the regional Environment Protection Agency (ARPA) in Aosta: Aosta–Saint Christophe, and Aosta–Downtown. The Aosta–Saint Christophe station (45.7°N, 7.4°E, 560 m a.s.l.) is located in a large floor with a wide field of view, at the bottom of the main valley, about 2.5 km east of Aosta–Downtown. The site is in a semi-

rural context, partially influenced by vehicular traffic and anthropogenic activities from the city, such as domestic heating and industry. The experimental setup at this site includes an Automated LiDAR-Ceilometer (ALC) for the operational monitoring of the aerosol profile (Dionisi et al., 2018), a POM-02 sun photometer for the retrieval of column aerosol properties and water vapour (Diémoz et al., 2014a; Campanelli et al., 2018), and a Fidas 200s Optical Particle Counter for the surface aerosol size distribution, in addition to instruments measuring solar radiation and trace gases (Diémoz et al., 2011, 2014b; Siani et al.,

2013, 2018; Federico et al., 2017). Aosta–Downtown (580 m a.s.l.) is a urban background site. This station is equipped with samplers for continuous monitoring of atmospheric pollution, mainly coming from car traffic, domestic heating and a steel mill located south of the city. To provide an idea of the aerosol load in Aosta–Downtown, the annual averages of $PM_{10}$ and $PM_{2.5}$ concentrations calculated for the last three years of measurements (2015–2017) range between 18–21 $\mu g\,m^{-3}$ and 11–12 $\mu g\,m^{-3}$, respectively. Despite these low average concentrations, daily $PM_{10}$ exceedance episodes with maxima up to about

100 $\mu g\,m^{-3}$ can be observed, their occurrence strongly depending on the encountered meteorological conditions (5 exceedance episodes in 2016, 13 in 2015 and 17 in 2017). Thus, there is the need to unravel their behaviour and the role played by regional transport from most polluted areas. Additional measurements used here were performed at the elevated sites of La Thuile (1640 m a.s.l.), Saint-Denis (840 m a.s.l.) and Antey (1040 m a.s.l.), and at Donnas, a low-altitude site (316 m a.s.l., Fig. 2b) close to the border with the Piedmont region, at the entrance of the Aosta Valley. La Thuile is a remote mountain site in a tributary valley

hosting a meteorological and air quality station managed by ARPA. Similarly, a weather station is operated in the village of Saint-Denis by the regional meteorological bureau. Antey is a further small village in a tributary valley where an ARPA mobile laboratory was temporary operated. Finally, the Donnas station is located in a rural area, only partially influenced by traffic and agricultural local activities, such as burning of agricultural residuals. However, due to its proximity to the Po basin, it is expected to be heavily influenced by pollution from the plain.

As the vertical dimension is important in this investigation, we also used measurements from an ALC operating in Milan (Fig. 2), this being representative of the contrasting conditions within the Po Valley. The system is located on the U9-building (45.5 °N, 9.2 °E, 132 m a.s.l.) of the University of Milano-Bicocca, in an urban background area northeast of the city centre. A full description of the site and measurements is reported in Ferrero et al. (2018).




## 3 Methods

### 3.1 Measurements

The experimental setup used in this work includes vertically-resolved measurements from ALCs (Sect. 3.1.1), vertically-integrated (columnar) aerosol measurements from both ground (Sect. 3.1.2) and space (Sect. 3.1.3), and in situ measurements

of aerosol concentration and composition (Sects. 3.1.4–3.1.5), complemented by ancillary gas-phase pollutants and meteorological measurements (Sect. 3.1.6). Table 1 summarises the instruments used throughout this study in their respective measuring stations.

### 3.1.1 Automated LiDAR Ceilometers

Vertical profiles of air constituents are particularly useful in identifying transport of pollutants of non-local origin. However,

the profiling capability of the Italian regional Environment Protection Agencies is still scarce. Over the Po basin, continuous monitoring of the atmospheric composition along the vertical profile is lacking and information at different altitudes is mostly available for short periods and during specific, dedicated field campaigns (e.g., Barnaba et al., 2007; Osborne et al., 2007; Raut and Chazette, 2009; Barnaba et al., 2010; Ferrero et al., 2014; Curci et al., 2015; Rosati et al., 2016; Bucci et al., 2018).

Light detection and ranging (LiDAR) instruments permit to resolve the vertical distribution of particles. The recent tech-

nological and data-processing advances (Wiegner and Geiß, 2012), and commercialisation, of simple LiDAR systems with operational capabilities allow to use this kind of system in monitoring (24/7) mode and in wide networks. In the present study, we employ two commercially-available ALCs (CHM15k-Nimbus, manufactured by Lufft GmbH, and formely by Jenoptik ESW), which have been operating since 2015 at the Aosta–Saint Christophe observatory and in Milan. Both ALCs are part of the Italian Alice-net (http://www.alice-net.eu/) and the European E-PROFILE (https://ceilometer.e-profile.eu/profileview)

networks. They allow for continuous vertical profiling of the radiation emitted by a single-wavelength (1064 nm) pulsed laser (Nd:YAG; 6.5–7 kHz; 8 µJ pulse$^{-1}$) and backscattered by the atmosphere. At the operating wavelength, the backscatter is mainly dominated by aerosols and clouds in the atmosphere, whereas interference by water vapour has been estimated to be negligible (Wiegner and Gasteiger, 2015). The systems enable a typical temporal resolution of 15 s (integration time) and a vertical resolution of 15 m, up to 15 km above the ground. Main limitations of the instruments are: 1) need for corrections in

the lowermost levels, and 2) blind view above thick clouds. 1) In the lowermost levels, the field of view (0.45 mrad) of the receiver is only partially overlapped to the laser beam (90% overlap is achieved at about 700 m), therefore, an overlapping-correction function is needed to correct the signal. This was provided by the manufacturer. 2) Thick clouds cause saturation in the detector signal (an avalanche photodiode operated in photocounting mode), followed by complete signal extinction. Thus, the attenuated backscatter above the cloud ceiling is not considered (nor plotted) in this study. The ALC firmwares used so far





**Table 1.** Observation sites, measurements and instruments employed in this study.

| Station | Elevation (m a.s.l.) | Measurement | Instrument | Data availability |
|---|---|---|---|---|
| Aosta–Saint Christophe (ARPA observatory) | 560 | Vertical profile of attenuated backscatter and derived products | CHM15k-Nimbus ceilometer | 2015–now |
| | | Aerosol columnar properties | POM-02 sun/sky radiometer | 2012–now[a] |
| | | Surface particle size distribution | Fidas 200s optical particle counter | 2016–now |
| Aosta–Saint Christophe (weather station) | 545 | Standard meteorological parameters | Siap and Micros | 1974–now |
| Aosta–Downtown | 580 | $PM_{10}$ hourly concentration | TEOM 1400a | 1997–now |
| | | $PM_{10}$ and $PM_{2.5}$ daily concentrations | Opsis SM200 | 2011–now |
| | | Water-soluble anion/cation analyses on $PM_{10}$ samples | Dionex Ion Chromatography System | 2017–now |
| | | EC/OC analyses on $PM_{10}$ samples | Sunset thermo-optical analyser | 2017–now[b] |
| | | NO and NO2 | Horiba APNA-370 | 1995–now |
| | | Standard meteorological parameters | Vaisala WA15 | 1995–now |
| South mountain slope | 550–1200 | Temperature profile | HOBO H8 Pro (10 thermometers) | 2006–now |
| La Thuile | 1640 | $PM_{10}$ hourly concentration | TEOM 1400a | 2015–now |
| | | NO and NO2 | Teledyne API200E | 1997–now |
| Saint-Denis | 840 | Standard meteorological parameters | Siap and Micros | 2002–now |
| Antey | 1040 | $PM_{10}$ daily concentration | Opsis SM200 | 2017 |
| Donnas | 316 | $PM_{10}$ daily concentration | Opsis SM200 | 2010–now |
| | | NO and NO2 | Teledyne API200E | 1995–now |
| Milan | 132 | Vertical profile of attenuated backscatter and derived products | CHM15k-Nimbus ceilometer | 2015–now |
| | | Standard meteorological parameters | Vaisala WXT5 | 2012–now |

[a] Underwent major maintenance in the second half of 2016 and January 2017.

[b] Available 4 days every 10.

(versions 0.730–0.743 for Aosta–Saint Christophe and 0.730 for Milan) provide the background-, overlap- and range-corrected attenuated backscatter (RCS) in terms of instrumental raw counts, i.e.

$$RCS(z,t) = \frac{(P(z,t) - B(t))\, z^2}{O(z)} \qquad (1)$$



where $P(z,t)$ is the signal intensity (raw counts) backscattered from a specific distance ($z$) and measured at ground, $B(t)$ the time-varying background baseline and $O(z)$ the overlap function. To express the backscatter coefficient in SI units and make the results comparable with other similar instruments, a calibration factor ($C_L$) must be assessed, so that

$$\frac{\text{RCS}(z,t)}{C_L} = \beta_{att}(z,t) = \beta_T(z,t)\,e^{-2\int_{z_{min}}^{z}\alpha_T(z',t)\mathrm{d}z'} \qquad (2)$$

where $\beta_{att}$ is the attenuated backscatter coefficient, $\beta_T$ is the total (particles and molecules) backscatter coefficient and $\alpha_T$ is the total extinction coefficient. $C_L$ is determined during clear-sky time windows of at least 3 hours at night, i.e. when the background radiation is low, by the method (Rayleigh technique) described hereafter. First, the backscatter and extinction profiles are calculated by the Klett-Fernald backward algorithm (Fernald, 1984; Klett, 1985), then $C_L$ is determined by inverting Eq. 2. Once a series of calibration factors has been estimated, the total ($\alpha_T$, $\beta_T$) and particle ($\alpha_p$, $\beta_p$) extinction and backscatter

coefficients are computed for all times and sky conditions using a forward Klett method as described by Wiegner and Geiß (2012).

Usually, the above-mentioned solving techniques are based on an *a-priori* or independent estimate of the lidar ratio (LR, i.e. the ratio $\alpha_p/\beta_p$) as a further constraint. In our case, LR is not fixed a-priori, but rather obtained using specific functional relationships linking $\alpha_p$ to $\beta_p$. Dionisi et al. (2018) demonstrated that this approach, previously proposed and tested on the

signal inversion of research-type elastic LiDARs (e.g., Barnaba and Gobbi, 2001, 2004), provides better retrievals of $\alpha_p$ and $\beta_p$ also from ALCs than using an a-priori, fixed LR. More specifically, an iterative data inversion scheme is adopted: at the first iteration, LR is set to an initial value of 38 sr (average value from the functional relationships) and a first retrieval of the backscatter coefficient $\beta_p$ is calculated; starting from the second iteration, the calculated backscatter coefficient and the functional relationships are used to determine an altitude-dependent lidar ratio. The loop continues until convergence of the

column-integrated backscatter is reached. The good agreement between the ALC-derived and the sun photometer-measured aerosol optical depth (AOD, i.e. the integral over altitude of the the extinction coefficient) is employed as a validation of the quality of the inversion results (Sects. 4.1.4 and 4.3.5) using these functional relationships, at least in daytime conditions (see also Dionisi et al. (2018) and Fig. S2d, Sect. S4, and Fig. S8e, Sect. S6, in the Supplementary material).

For ease of comparison with pristine (aerosol-free) conditions, and with most LiDAR-based studies, ALCs measurements

are provided in this study in terms of scattering ratio (SR, e.g. Zuev et al., 2017), i.e.

$$\text{SR} = \frac{\beta_T}{\beta_m} = \frac{\beta_p + \beta_m}{\beta_m} \qquad (3)$$

where $\beta_m$ is the molecular backscatter coefficient. In case of pure molecular scattering (no aerosol in the atmosphere), SR $= 1$, while SR increases with increasing aerosol load. Finally, the high-resolution data from the ALC are downscaled to 75 m averages over the vertical and 5 min averages over time to increase the signal-to-noise ratio. An example of the output from

the Aosta–Saint Christophe ALC, in terms of scattering ratio, is depicted in Fig. 3. The image refers to a typical advection day (25 May 2017), characterised by a relatively clean atmosphere during the morning and a sudden increase of the particle





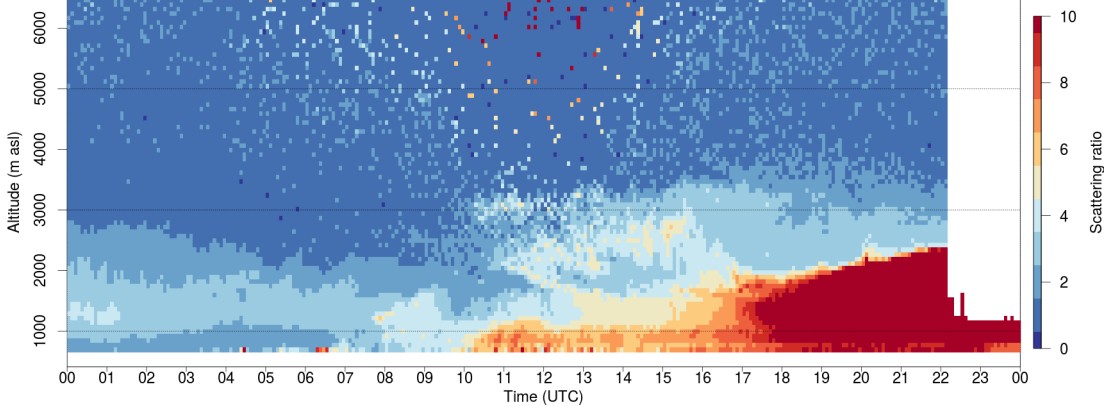

**Figure 3.** Example of particle backscatter profile, in terms of scattering ratio (SR), from the ALC in a typical advection day (25 May 2017, cf. Fig. 13). Areas above the detected cloud ceiling, where the backscatter from the ALC is not reliable, are plotted in white. Noisy measurements (e.g. interference by solar radiation in the middle of the day) are filtered on the basis of a spatial and temporal variability criterion (the standard deviation for each bin not exceeding 20% of the mean value). The Aosta–Saint Christophe ALC is located at an altitude of 560 m a.s.l.

backscatter during the afternoon, up to an altitude of more than 2000 m a.s.l. (the altitude of the surface being 560 m a.s.l in Aosta–Saint Christophe). As we will demonstrate here, this afternoon increase is due to the transport of polluted air masses from the Po basin (see Sect. 4). Several analogous episodes were recorded in the ALC record since its installation in Aosta–Saint Christophe. The observation of this recurrent phenomenon was, in fact, the driving motivation for the present research.

5  In the study, we also convert the ALC-derived backscatter into aerosol volume following Dionisi et al. (2018), thus allowing a direct comparison to more standard air quality metrics (e.g. $PM_{10}$, this is done using a particle density $\rho = 1.3\,\mathrm{g\,cm^{-3}}$ independently estimated for the present study by an optical particle counter co-located with the ALC). The expected uncertainties in the retrieval of the aerosol backscatter and extinction coefficients, and of the aerosol volume range between 30 and 40% (Dionisi et al., 2018).

### 3.1.2 Sun photometer

A POM-02 sun/sky radiometer operates at the Aosta–Saint Christophe observatory since 2012. The radiometer is part of the European ESR-SKYNET network (http://www.euroskyrad.net/). The irradiances collected by the POM-02 at 11 wavelengths (315–2200 nm) are inverted to retrieve the aerosol optical properties using both the direct-sun (SUNRAD.pack algorithm to provide the AOD every 1 min, Estellés et al. (2012)) and the almucantar geometries (SKYRAD.pack software version
15 4.2 to retrieve a complete set of optical and microphysical columnar properties every 10 min, Nakajima et al. (1996)). The instrument is calibrated in-situ with the improved Langley technique, described by Campanelli et al. (2007) in more detail, and



was successfully compared to other reference instruments during a recent international campaign (Kazadzis et al., 2018). The AOD ($\tau$) from the POM-02 is interpolated to the ALC wavelength (1064 nm) using the Ångström (1929) relationship, i.e.

$$\tau = b\lambda^{-a} \tag{4}$$

where $\lambda$ is the wavelength expressed in µm and $a$ and $b$ are the Ångström parameters from the regression. Finally, the Cloud
Screening of Sky Radiometer data (CSSR) algorithm by Khatri and Takamura (2009), making use of the short-wave irradiance measurements by a co-located pyranometer, is applied to the POM-02 series to minimise the residual interference by clouds and to ensure the maximum measurement quality.

### 3.1.3   Space-based observations from MODIS

In this study, we also used satellite data to explore if and how the "local" phenomenon observed in Aosta is detectable over
a regional scale. To this purpose, we used AOD data from the Moderate Resolution Imaging Spectroradiometer (MODIS) instrument. The MODIS instrument flies onboard the two NASA platforms Terra and Aqua, following a sun synchronous orbit with overpass time between 10.00 and 13.00, and 13.00 and 16.00 (local time), respectively. Since the MODIS instrument planning phase, specific retrievals have been set up to provide the AOD over ocean and land globally on a daily base at 10 km resolution (Kaufman and Tanré, 1998). Constant improvements to the AOD inversion algorithms currently allow to provide
a 3 km-resolved standard AOD product (Remer et al., 2013). While such spatial resolutions have been extensively exploited for many regional-scale, aerosol-related studies, these are yet not sufficient for applications requiring more spatial detail, as in space-based evaluations of air quality within urban areas (e.g., Chudnovsky et al., 2014; Della Ceca et al., 2018) or in conditions of high AOD spatial variability as over mountain regions (e.g., Emili et al., 2011). For our purpose, we therefore used high-resolution (1 km) AOD data obtained inverting MODIS data with the recently developed algorithm MAIAC (Multi-
Angle Implementation of Atmospheric Correction). Full details of this algorithm are thoroughly described in Lyapustin et al. (2011, 2012).

### 3.1.4   Optical Particle Counter

A Fidas®200s (Pletscher et al., 2016) optical particle counter operates at the ARPA observatory in Aosta–Saint Christophe. The spectrometer is based on the analysis of scattered light at 90° originating from a polychromatic light source (LED).
These conditions ensure an accurate calibration curve without ambiguities within the Mie range and allow to retrieve high-resolution spectra (size measurements between 0.18 and 18 µm, 32 channels/decade). Due to the peculiar T-aperture optics of the spectrometer and the simultaneous measurement of signal duration, border zone errors are eliminated. Once the particle size distribution is measured, the instrument algorithm is able to derive the mass concentration for several cutoff diameters (including $PM_{10}$, i.e. ambient particulate with a diameter of 10 µm or less). Though not a direct mass measurement, the PM
concentration derived by the instrument obtained the certificate of equivalence to the gravimetric method by TÜV Rheinland Energy GmbH on the basis of a laboratory and a field test. Moreover, to prevent any site-specific bias, an additional $PM_{10}$





comparison with the gravimetric technique was organised at Aosta–Saint Christophe and provided satisfactory results (29 days; slope $1.08 \pm 0.04$; intercept $-3.8 \pm 1.4\ \mu g\,m^{-3}$; $R^2 = 0.96$).

### 3.1.5   PM concentration and composition

Daily averages of PM concentration are recorded by four Opsis SM200 Particulate Monitor instruments, two in Aosta–
Downtown ($PM_{10}$ and $PM_{2.5}$ inlets, with sampling fluxes of $1\ m^3\,h^{-1}$ and $2.3\ m^3\,h^{-1}$, respectively), one installed inside a mobile laboratory, which was parked in Antey ($PM_{10}$, $1\ m^3\,h^{-1}$), and one in Donnas ($PM_{10}$, $1\ m^3\,h^{-1}$). Moreover, two Tapered Element Oscillating Microbalance (TEOM) 1400a monitors (Patashnick and Rupprecht, 1991) are used for continuous measurements of $PM_{10}$ hourly concentrations at the stations of Aosta–Downtown and La Thuile. These instruments are not compensated for mass loss of semi-volatile compounds (Green et al., 2009), therefore they are only employed for qualitative
estimates of short-term variations of the aerosol burden and could be insensitive to specific compounds, such as ammonium nitrate (e.g., Charron et al., 2004).

Sampling in Aosta–Downtown is complemented with chemical speciation analyses. We employed a Dionex Ion Chromatography System (AQUION/ICS-1000 modules) for water-soluble anion/cation chemical analyses on daily $PM_{10}$ samples collected on PTFE-coated glass fiber filters by the Opsis SM200. The experimental setup is based on the CEN/TR 16269:2011
guideline and enables the determination of mass concentrations of the following water-soluble ionic compounds: $Cl^-$, $NO_3^-$, $SO_4^{2-}$, $Na^+$, $NH_4^+$, $K^+$, $Mg^{2+}$, $Ca^{2+}$. Samples collected on quartz fibre filters by a co-located Micro-PNS automatic low-volume sampling system (10 μm cutoff diameter, $2.3\ m^3\,h^{-1}$) are analysed alternatively for elemental/organic carbon (EC/OC, 4/10 days) and for metals (6/10 days, not used in the present study, but discussed by Diémoz et al. (2018)). The carbonaceous aerosol mass is determined with a Sunset Laboratory Inc. instrument (Birch and Cary, 1996) on portions of $1\ cm^2$ punches us-
ing a thermal-optical transmission (TOT) method with transmission correction for the split point and following the EUSAAR-2 protocol (Cavalli et al., 2010), according to the EN 16909:2017.

### 3.1.6   Gas-phase pollutants and meteorological ancillary data

Standard gas-phase pollutants subject to European regulations are routinely monitored at Aosta–Downtown, La Thuile and Donnas in the frame of the activities of the air quality network. Meteorological parameters, such as temperature, pressure,
relative humidity (RH) and surface wind velocity are collected at the stations of Aosta–Saint Christophe and Saint-Denis. Moreover, 10 temperature sensors (Hobo H8 Pro) are installed along the north-facing mountain slope south of Aosta, at elevations ranging from 550 to 1200 m a.s.l. This set of measurements, representing a vertical profile of surface temperatures, provides useful information about the thermal inversions in the main valley.

## 3.2   Models

Models are used to interpret and complement the observations. A numerical weather prediction model (COSMO, Consortium for Small-scale Modeling, www.cosmo-model.org, Sect. 3.2.1) is employed to drive a chemical transport model (FARM,



Flexible Air quality Regional Model, Sect. 3.2.2) and a lagrangian model (LAGRANTO) to retrieve the trajectories of air masses arriving at the experimental site (Sect. 3.2.3).

### 3.2.1 Numerical Weather Prediction model

COSMO is a non-hydrostatic, fully compressible atmospheric prediction model working on the meso-$\beta$ and meso-$\gamma$ scales. A detailed description of the model can be found elsewhere (e.g., Baldauf et al. (2011)). The COSMO data are operationally disseminated by the meteorological operative centre – air force meteorological service (COMET) in two different configurations: a lower-resolution (7 km horizontal grid and 45 levels vertical grid, 72 hours integration) version (COSMO-ME), covering the central and southern Europe, and a nudged, higher-resolution (2.8 km, 65 vertical levels, 4 runs/day), called COSMO-I2 (or COSMO-IT), covering Italy (Fig. 2). Owing to the complex topography of the Aosta Valley and the consequent need to resolve the atmospheric circulation at very small spatial scales, the COSMO-I2 variant is employed in this work.

As an example of the good agreement between COSMO and surface measurements, the average daily cycle of the wind speed and direction from both data sources is exhibited in Fig. S1, Sect. S1 in the Supplement. The figure clearly shows the regular development of the plain-mountain winds in the afternoon. The influence of the east-west directrix of the main valley along which the wind is channelled is well represented in both measurements and simulations.

### 3.2.2 Chemical Transport model

FARM (v4.7, Gariazzo et al., 2007; Silibello et al., 2008; Cesaroni et al., 2013; Calori et al., 2014) is a three-dimensional Eulerian model for simulating the transport, chemical conversion and deposition of atmospheric pollutants. The FARM source code has been inherited from the Sulfur Transport and dEposition Model (STEM), extensively tested and used since the eighties.

FARM can be easily interfaced to most available diagnostic or prognostic NWP models. A turbulence and deposition pre-processor (SURFace-atmosphere interface PROcessor, SURFPRO) computes the 3-D fields of turbulence scaling parameters, eddy diffusivities and deposition velocities for each species based on an input gridded land-use field and the results of the NWP model (Sokhi et al., 2003). Pollutants emission from both area and point sources can be simulated by FARM including plume rise calculations. Transformation of chemical species by gas-phase chemistry (more than 200 reactions using the SAPRC-99 chemical scheme as in Carter (2000)), dry removal of pollutants depending on local meteorology and land-use, and wet removal are considered. The AERO3_NEW module, coupled with the gas-phase chemical model and treating primary and secondary particle dynamics and their interactions with gas-phase species, is implemented for the calculation of the aerosol concentration fields, thus accounting for nucleation, condensational growth and coagulation (Binkowski, 1999). The aerosol size distribution is parametrised using three modes simulated independently: the Aitken mode (D < 0.1 µm), the accumulation mode (0.1 µm < D < 2.5 µm) and the coarse mode (D > 2.5 µm). $PM_{2.5}$ is defined as the sum of Aitken and accumulation modes, while $PM_{10}$ is given by the sum of the three modes. Chemical speciation is performed in the pre-processing phase by the emission manager (EMMA) based on the profiles from the US EPA model SPECIATE (v3.2, 2002; cf. https://www.epa.gov/air-emissions-modeling/speciate-version-45-through-40 for more recent versions). To simulate hygroscopic growth by aerosols in high relative humidity conditions, water uptake by aerosol particles is taken into account based on the ISORROPIA





model (Nenes et al., 1998) and added to the $PM_{10}$ dry mass concentration from FARM. The resulting output species is called $PM_{10w}$ in FARM version 4.7.

The FARM output concentrations are 4-D fields at 1 km spatial resolution along the horizontal dimensions, 16 different vertical levels (from the surface to 9290 m, corresponding to equispaced pressure levels) and 1-hour temporal resolution.

The $PM_{10w}$ concentration profiles from FARM are extracted at the grid cell corresponding to Aosta–Saint Christophe for comparison with the profiles measured by the ALC. Indeed, since FARM is not able to calculate the aerosol optical properties needed to simulate the backscatter coefficient measured by the ALC, the comparison between the profiles measured by the ALC and estimated by the CTM is here performed in terms of mass concentration (by converting the ALC data into $PM_{10}$, see Sect. 3.1.1).

Supplying a detailed and precise emission inventory to the CTM is crucial to accurately assess the magnitude of the pollutants loads and their variability in both time and space. Additional information regarding the regional emission inventory and the boundary conditions is provided in the Supplement (Sect. S2 and S3). The geographic coverage of the regional and the national emission inventories is shown in Fig. 2. Local sources and boundary conditions can be switched on/off for sensitivity analyses.

### 3.2.3 Back-trajectories

The publicly-available LAGRANTO Lagrangian analysis tool, version 2.0 (Sprenger and Wernli, 2015), is used to numerically integrate the high-resolution 3-D wind fields from COSMO and to determine the origin of the air masses sampled by the ALC over Aosta–Saint Christophe. The software also enables to trace 3-D and 2-D meteorological fields along each trajectory. In particular, the algorithm was set up to start 8 trajectories in a circle of 1 km around the observing site and at 7 different altitudes from the ground to 4000 m a.s.l., for a total of 56 trajectories for every run. From a one-year (2016) analysis of the trajectories

arriving to the Aosta Valley, it is found that a backward run time of 48 hours is sufficient, on average, to cover most of the domain of the meteorological model. Therefore, to reduce errors in trajectories with increasing running time, we limit the computation to this duration.

## 4 Results

The observed phenomena are presented through three case studies (26–31 August, 2015; 26–29 January, 2017; 25–30 May,

2017), chosen for their relevance and clarity. The episodes are also representative of three different atmospheric conditions (seasons) and were observed with slightly different sets of operating instruments (Table 1). The case studies were also selected among those showing long sequences of days characterised by the recurrent appearance of a thick aerosol layer from the ALC, to emphasise the periodicity of the phenomenon. Indeed, as explained in more detail by Diémoz et al. (2018), the elevated aerosol layer can be observed very frequently, i.e. at least 40–50% of the days, depending on the season.




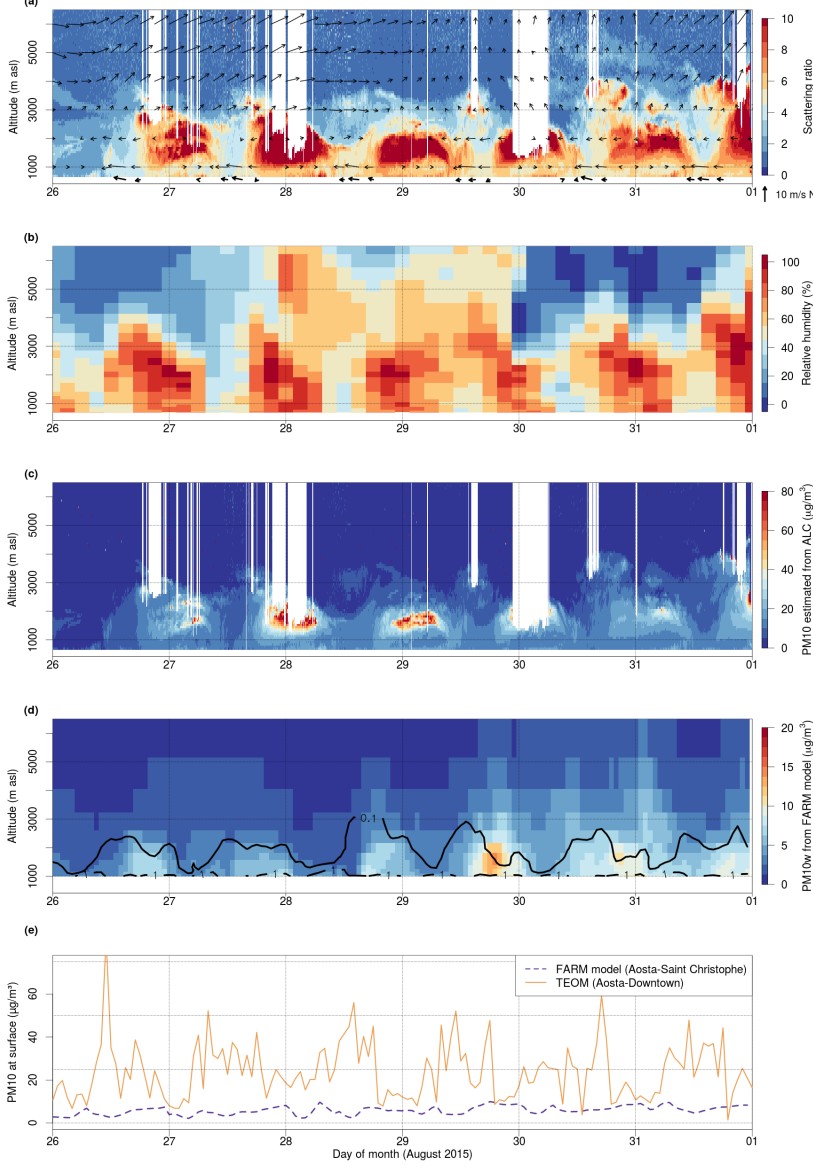

**Figure 4.** Case study of 26–31 August 2015. (a) Coloured background: vertical profile of scattering ratio from ALC in Aosta–Saint Christophe. The signal above the clouds is plotted as white areas. Arrows: horizontal velocity of the wind measured at the surface (bold, lower arrows) and simulated by COSMO at several elevations (thin arrows). Calm wind (speed $< 1 \, \mathrm{m \, s^{-1}}$) is not plotted. A reference arrow for a $10 \, \mathrm{m \, s^{-1}}$ wind blowing from the south to the north is drawn at the bottom right corner; (b) Vertical profile of relative humidity forecasted by COSMO; (c) Vertical profile of $PM_{10}$ mass concentration derived from ALC; (d) Mass concentration ($PM_{10w}$) from FARM. PM concentration from non-local sources is represented by the coloured background (the colour scale is chosen to better show the daily pattern simulated by the model) and the effect of local sources by the contour line, at logarithmic steps; (e) Hourly $PM_{10}$ (dry) surface concentration from FARM simulations in Aosta–Saint Christophe and observations in Aosta–Downtown, for the purpose of checking if any sudden variation in surface air quality data is noticeable.




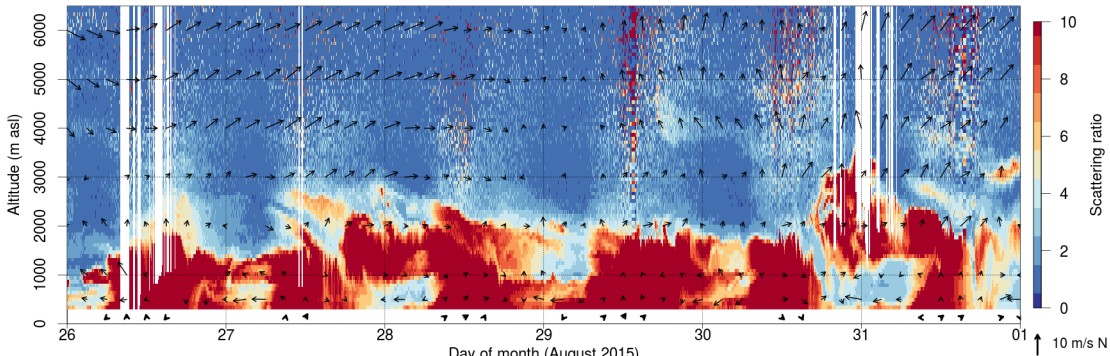

**Figure 5.** Vertical profile of scattering ratio from the ALC in Milan. Arrows: horizontal velocity of the wind measured at the surface (bold, lower arrows) and simulated by COSMO at several elevations (thin arrows).

## 4.1 Case study 1: Summer (26–31 August 2015)

One of the longest and most notable episodes of unexpected high aerosol loads in the northwestern Alps was registered from 26 August to 3 September 2015 (here we present the period 26–31 August), few months after the ALC installation in Aosta–Saint Christophe. In those days, a wide anticyclonic area extended from northern Africa to central and eastern Europe. The period is thus representative of fair weather conditions, with only few cirrus clouds on days 27 and 28, and absence of strong synoptic flows at ground, which favoured the regular development of thermally-driven winds from the plain to the mountains triggered by temperature and pressure gradients between the valley and the foreground.

### 4.1.1 ALC observations

A thick aerosol layer is detected by the ALC over the Aosta–Saint Christophe observatory from the afternoon of 26 August (Fig. 4a). The appearance of the layer is clearly noticeable on this day as an increase in the backscatter coefficient, with scattering ratios, SR $\simeq$ 4 at midday (light-blue area in the figure) almost doubling (SR$>$ 8, red) in few hours. This layer persists during the night, when SR reaches values above 30. The day after, when convection starts in the valley after sunrise, the aerosol-rich layer is observed to entrain into the developing mixing layer, thus impacting the lower levels, with potential consequences on the surface air quality, as previously observed in other areas (e.g., Bader and Whiteman, 1989; Curci et al., 2015). On 27 August, the ALC backscatter is then observed to decrease in the central part of the day and to increase again in the afternoon. This behaviour keeps very regular for almost a week, with the afternoon aerosol-rich layer extending from ground up to 3–3.5 km. At night-time, some low clouds form within the aerosol layer and are screened out in the figure (white areas). The transition from aerosol to the cloud phase is very sharp, as noticeable from the sudden increase of more than 40 W m$^{-2}$ of the downward infrared irradiance monitored at the same site (Fig.S2c).





Simultaneous ALC measurements in Milan (see relative position in Fig. 2), which can be considered representative of the overall dynamics occurring in the Po basin, are shown in Fig. 5. Interesting feature here is that the modulation of the scattering ratio looks almost reversed compared to Aosta–Saint Christophe, with maximum SR at the surface at midday and minimum values during the night and the morning. While in the uppermost levels (>3000 m a.s.l.) the synoptic circulation is blowing undisturbed from the west, the wind velocity at 500 m a.s.l. keeps alternating, likely driven by the breeze regime (the surface wind is affected by urban effects and does not show appreciable variations).

### 4.1.2 Meteorological variables and back-trajectories

The observed reversal behaviour in Milan and Aosta already suggests that air masses movements are driving the clean-up of the lowermost levels in the Po plain and transporting the aerosol plumes elsewhere. To substantiate this hypothesis, a careful analysis of the meteorological fields (observed and modelled) was performed. In particular, we verified that this selected sequence of days presents a typical pattern of plain-to-mountain wind systems during the afternoon of each day in Aosta–Saint Christophe. Surface-level eastern winds speed as high as $8 \, \mathrm{m \, s^{-1}}$ is measured daily in the afternoon till sunset and is shown as bold arrows in the lowermost levels of Fig. 4a. Conversely, calm wind is detected during the night, i.e. when the aerosol layer thickens. Since no instrument is available at the measuring site to determine the vertical profile of the wind velocity, the simulations from the COSMO model are used to assess the wind field at several altitudes (thin arrows in Fig. 4a). It reproduces well the thermal wind circulation in the lowest atmospheric layers during the afternoon and slightly overestimates the mountain-to-plain drainage winds at night and early morning. The thermally-driven wind pattern forecasted by COSMO extends up to an altitude of $3000 \, \mathrm{m}$, i.e. approximately the maximum height of the aerosol layer observed by the ALC. Note that wind direction is incompatible with the Aosta city being the potential source of the observed aerosol layer, as it is located west of the observatory. At higher elevations, the wind field is clearly decoupled from that in the PBL and follows the large-scale circulation.

Complementary information is provided by the analysis of the 48-hours back-trajectories calculated by LAGRANTO using COSMO fields (Sect. 3.2.3) ending over the Aosta Valley in the period addressed (Fig. 6 and S3). These results show that during the night between 25 and 26 August trajectories are driven by large-scale flows from the northwest direction and are thus parallel at all altitudes. This indicates air masses reaching Aosta to have crossed the Alps, notably the Mt Blanc chain, before arriving over the observatory, hence transporting clear and unpolluted air from the free troposphere to the PBL. Then, in the morning of 26 August from 9 to 12 UTC, back-trajectories in the PBL suddenly change their provenance owing to the development of the thermal circulation tapping into air masses of very different origin. The lowermost trajectories cover a notable distance and cross some major conurbations of the Po basin, i.e. Milan and Turin, at altitudes lower than few hundreds meters a.s.l., and thus well within the polluted PBL. The sudden reversal of the trajectories occurs simultaneously with the appearance of the elevated aerosol layers in the Aosta ALC image (Fig. 4a). These meteorological conditions persist for the rest of the day and in the following days. The analysis of the corresponding back-trajectories confirms that the air masses sampled by the ALC in Aosta–Saint Christophe keep originating from the Po basin for the whole episode.




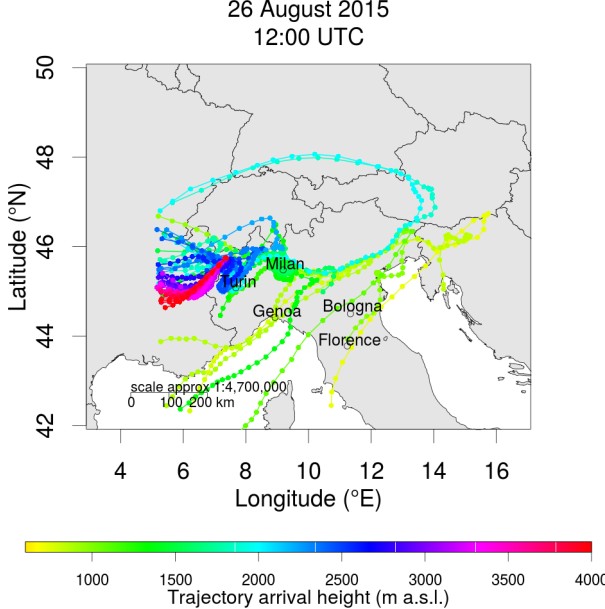

**Figure 6.** 48-hours back-trajectories ending at Aosta–Saint Christophe on 26 August 2015 at 12 UTC. The trajectories are cut at the border of the COSMO model. The colour scale represents the arrival height. The dots along each trajectory mark a 1-hour step. The whole sequence for the day 26 August is shown in Fig. S3, Sect. S4.

To complete the picture, it is worth mentioning that the COSMO model also predicts an increase of the relative humidity (Fig. 4b) from evening to morning, almost simultaneous with the SR enhancement observed by the ALC. In this time frame, RH exceeds typical summertime deliquescence values reported for the Po basin in previous studies (e.g., DRH=67%, D'Angelo et al., 2016), and reaches up to 98% at the ground (Fig. S2b). This suggests hygroscopic growth on aerosols and consequent

increase in the ALC $\beta_p$ (e.g., in a measurement site representative of Po Valley conditions, Adam et al. (2012) found a median increase of the aerosol backscatter coefficient of 70% for RH=90% compared to the dry case). During the day, RH decreases below typical crystallisation values in summer (e.g., CRH=62%, D'Angelo et al., 2016). As RH is clearly modulated by the temperature daily cycle, the measured specific humidity (SH) is also plotted on the same figure (S2b) as an additional variable, independent of temperature, to identify potential advections of different air masses to the observation site. Indeed, an SH

increase occurs on the first day of the sequence (starting from minimum values of ∼8 $\mathrm{g\,kg^{-1}}$ in the morning to about 11 $\mathrm{g\,kg^{-1}}$ in the evening) as soon as the wind starts blowing and high values ($> 13\,\mathrm{g\,kg^{-1}}$) endure for the rest of the week, likely indicating that the dry air, typical of the more mixed mountain PBL (Henne et al., 2005; Mélin and Zibordi, 2005), is replaced by more stagnating, and humidified, air masses characteristic of hot summer days in the Po Valley (Bucci et al., 2018). This scenario is compatible with recent findings by Campanelli et al. (2018), who performed water vapour measurements with the

POM-02 at Aosta–Saint Christophe and found that moist air masses are mainly coming from the east.





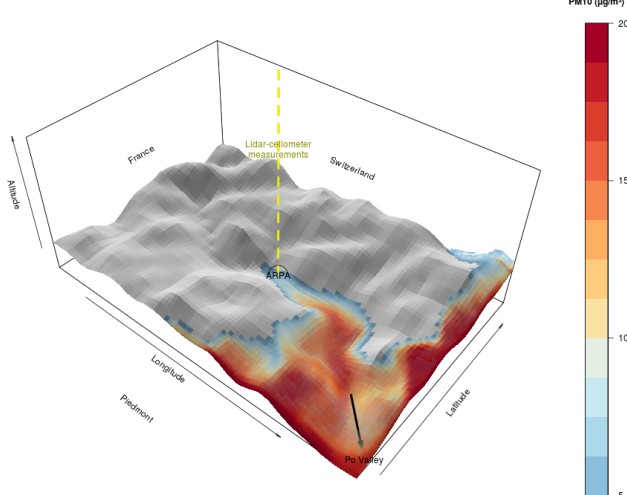

**Figure 7.** Still-frame of the three-dimensional simulation of $PM_{10}$ concentration by FARM (image from 28 August 2015 at 15 UTC). The image clearly shows the entrance of the aerosol-rich air mass from the Po basin into the Aosta Valley (red and blue area). The same colour scale as in Figs. 4d, 10d and 13d is used (the lowest concentrations are removed for ease of representation). The sequence 26–31 August 2015 is available as a video file in the Supplementary material.

### 4.1.3 Mass concentrations

We show in Fig. 4c the altitude-resolved aerosol mass derived from the ALC backscatter coefficient (as described in Sect. 3.1.1). The maximum concentration within the aerosol layer is $> 80\,\mu g\,m^{-3}$. The corresponding $PM_{10w}$ profile from FARM, partitioned between the non-local (coloured background) and local (contour line) pollution, is shown in Fig. 4d. FARM qualitatively reproduces the recurrent increase of the aerosol concentration at the end of each day and mainly ascribes it to particles transported by the thermal winds from the model-box boundaries. As an example, Fig. 7 provides a 3-D snapshot of the model simulation results, clearly showing the entrance of the aerosol-rich air mass from the Po basin to the Aosta Valley. The picture refers to 28 August 2015 at 15 UTC – the whole sequence 26–31 August 2015 being available as a video file in the Supplementary material. Still, there are two important differences between the FARM model simulations and the ALC observations in terms of 1) absolute $PM_{10w}$ concentrations and 2) timing of the phenomenon. In fact:

1. $PM_{10w}$ values from FARM are much lower than the ones retrieved from the ALC (about -40% outside the thick aerosol layer identified by the ALC at night and even -80% inside the layer);

2. the maximum $PM_{10w}$ simulated concentration during the advections is anticipated by several hours (up to 6–7 hours, in the worst cases) compared to the ALC measurements, which, in contrast, show a better correlation with the relative humidity profile by COSMO.





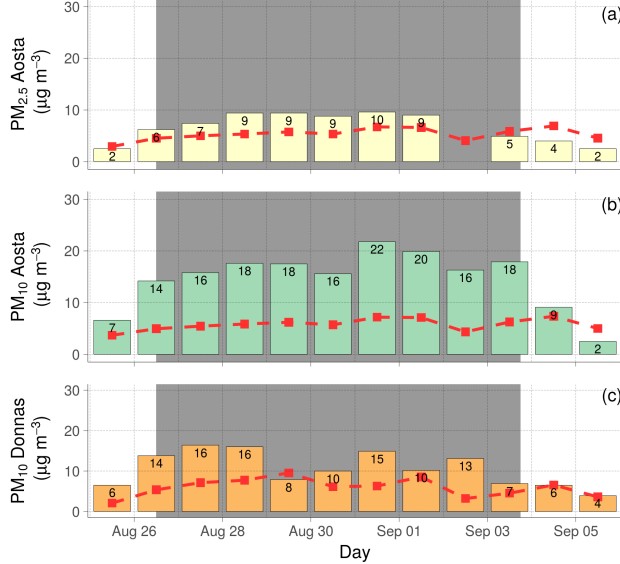

**Figure 8.** Measured (coloured bars) and simulated (dotted line) daily averages of $PM_{2.5}$ and $PM_{10}$ concentrations at Aosta–Downtown and Donnas during case study 1 (August 2015). The period when the ALC detects a thick layer above Aosta–Saint Christophe is highlighted with a grey background. $PM_{2.5}$ measurements in Aosta are missing for 2 September 2015.

Possible reasons, such as hygroscopicity effects and modelling deficiencies, explaining the above-mentioned issues are further discussed in Sect. 4.4.

To evaluate the impacts on surface air quality parameters during the episode, hourly $PM_{10}$ concentrations at the surface as measured in Aosta–Downtown and simulated by FARM in Aosta–Saint Christophe are presented in Fig. 4e ($PM_{10}$ monitoring

at La Thuile was not yet operational, at that time). Apart from one spike (80 $\mu g\,m^{-3}$) on 26 August of local origin, the concentrations measured in Aosta–Downtown are generally higher during daytime compared to the night and do not show any noticeable increase corresponding to the arrival of the layer. This feature, however, can be connected to the fact that mass loss occurs in TEOM due to secondary aerosol volatility, as also found by Diémoz et al. (2018) by comparing the daily $PM_{10}$ cycle from this instrument and the Fidas OPC in Aosta–Saint Christophe. Besides, FARM estimates at the surface are again lower

than measurements (-60%, on average). Daily $PM_{10}$ concentrations observed by Opsis SM200 instruments during the case study in Aosta–Downtown and Donnas are shown in Fig. 8, the shaded area corresponding to those dates affected by the thick layers as revealed by the ALC. Unlike hourly measurements by TEOM, an increase in daily concentrations (up to 7 $\mu g\,m^{-3}$ for $PM_{2.5}$ and 11–15 $\mu g\,m^{-3}$ for $PM_{10}$) can be clearly noticed at both sites. The daily averages of the simulated aerosol concentrations at the surface are superimposed on the same figure (dashed lines). While the model qualitatively reproduces the

average load of $PM_{2.5}$ and its variations in Aosta–Downtown, it underestimates $PM_{10}$ at both stations as already noticed.



### 4.1.4 Sun photometer measurements

Since sun photometric measurements can be only performed in daylight, results are often unavailable at those times when the ALC shows the greatest backscatter signal, i.e. in the evening and at night. However, data collected by the POM-02 radiometer can still be effective to monitor the first (late afternoon) and last (early morning) dynamics of the aerosol layer as seen by the ALC (Fig. 4a), and particularly tell us if this signal is detectable in the sun photometer-derived, column-integrated aerosol load. AOD obtained from the sun photometer (Fig. S2d) varies from 0.02 (26 August, before appearance of the layer) to 0.07 (29 August, morning) at 1064 nm (approximately 0.05 to 0.2 at 500 nm) and closely follows the AOD obtained by vertically integrating the extinction coefficient from the ALC over the atmospheric column. The two independent AOD retrievals present a mean bias of -0.007 and standard deviation of the differences of 0.006, both lower than the declared uncertainty of the POM sun photometer itself (about 0.01) (Campanelli et al., 2007). The good closure with the AOD from the photometer demonstrates the reliability of the functional relationships derived by Dionisi et al. (2018) and employed in our ALC inversion algorithm, at least during the daytime.

Further retrieval products from SUNRAD.pack and SKYRAD.pack (displayed in Fig. S2e) show the Ångström exponent to increase from 1.2 to 1.7 on the first day from 8 to 17 UTC, suggesting the advection of smaller particles in the atmosphere, and to remain almost constant (about 1.6, a typical value for the Po Valley, as already described by Mélin and Zibordi (2005), and Kambezidis and Kaskaoutis (2008)) in the following days. Likewise, the single scattering albedo (SSA) at 500 nm increases (from 0.7 to 0.95) on 26 August, which is compatible with the arrival of more scattering (likely secondary aerosol, as described in Sects. 4.2.4 and 4.3.4) and/or more aged aerosol, such as that from the Po Valley (Barnaba et al., 2007; Gilardoni et al., 2014). The sun photometer-derived, total-column, aerosol volume distribution (Fig. S2f) peaks in the accumulation mode (about 0.3 μm). A slight decrease of the peak diameter in the morning (from about 0.4 μm to 0.2 μm) can be noticed on some days (e.g., 27–30 August) and might be ascribed to the dehydration of the particles as temperature increases and RH decreases. The same behaviour can be observed better in the third case study (Sect. 4.3.5).

### 4.1.5 Spatial extent of the observed phenomenon

In order to provide a first evaluation of whether the phenomenon observed and described in detail for the Aosta area could have a more general validity in the Alpine region, we used AOD data retrieved from space over Northern Italy. In particular, we exploited the high resolution capabilities of the MODIS-MAIAC AOD product (Sect. 3.1.3) and the availability of two MODIS overpasses during the day (Terra and Aqua platforms), to detect signs of the described effects at the regional scale. Figure 9a shows the average difference between the AOD retrieved each day from MODIS-Aqua (overpass time between 12–13 UTC) and that from MODIS-Terra (10-11 UTC). Despite the short time lag between the Terra (AM) and Aqua (PM) satellite overpasses, this figure shows that the data are sufficient to start detecting an overall reduction of the AOD in the Po basin (blue area) and a reverse increase in the mountain areas (Alps and Apennines) surrounding it. The general picture suggests a sort of aerosol drainage from the Po Valley (negative AOD difference, blue) to the Alps (positive AOD difference, red), although some aerosol dehydration from the morning to the afternoon could also partially contribute to the observed PM–AM differences. The





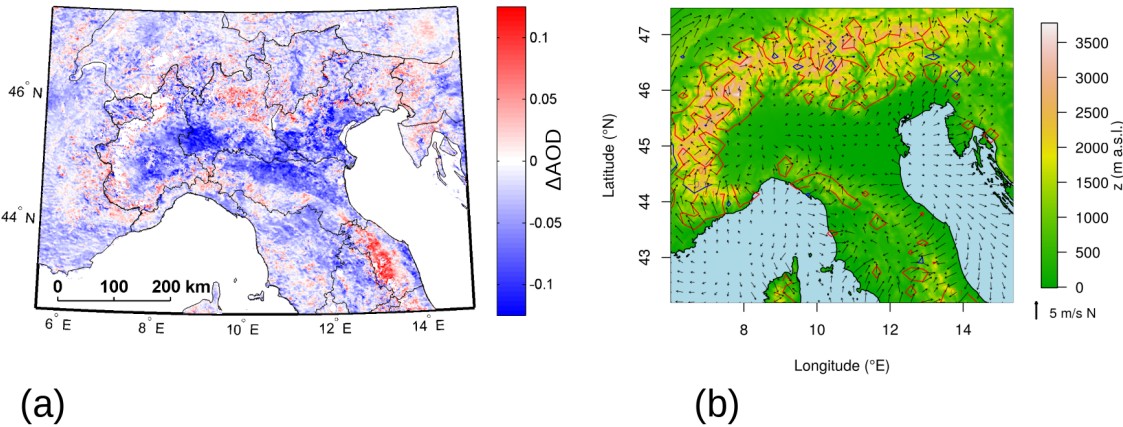

(a)   (b)

**Figure 9.** (a) Average difference between AOD estimated from Aqua and Terra satellites during days 27–31 August 2015 using the MAIAC algorithm. (b) Horizontal wind velocity from COSMO (arrows), vertical velocity (red/blue contours, $\pm 0.1$ ms$^{-1}$) over the same domain and the same hours as in Fig. (a).

hypothesis of aerosol transport, in agreement with our previous results from FARM (e.g., Fig. 7 and the relative video file), is strengthened by the wind simulations from COSMO over the same area and averaged over the same hours between Terra and Aqua overpasses (Fig 9b). Valley-mountain (and sea-land breezes) are clearly reproduced, as expected in days with weak synoptic flows and strong heating by the sun.

## 4.2 Case study 2: Winter (26–29 January 2017)

A second pollution transport episode was chosen for its significance and its consequences on air quality. Indeed, the last days of January 2017 and the first ones of February 2017 were characterised by heavy exceedances of PM$_{10}$ in the whole Po basin with concentrations of nearly 300 µg m$^{-3}$ in some stations of northern Italy (Bacco et al., 2017). This situation was driven by conditions of strong atmospheric stability, weak winds, low mixing height and presence of clouds, and additionally worsened by the transit of a warmer air mass aloft, i.e. the typical circumstances causing the most severe air pollution episodes in the Po basin in winter (Finardi and Pellegrini, 2004). Chemical analyses accomplished in the framework of the air quality monitoring network in northern Italy identified considerable formation of secondary particulate (e.g., ammonium nitrate), also confirmed by very large PM$_{2.5}$/PM$_{10}$ ratios (almost 90%).

In the Aosta Valley, this pollution episode lasted only from 26 to 29 January. At that time, the Alps were contended by a pressure trough at the north and a ridge at the south. At the beginning of the period, the influence of the low-pressure system prevailed and brought cloudy skies over the valley, thus enforcing the atmospheric stability in the PBL. The PM concentrations measured in the Aosta Valley, although lower than the ones detected in the Po basin, were found to be significant in the whole region (e.g., PM$_{10}$>100 µg m$^{-3}$ in Aosta–Downtown and Donnas), even at some remote measuring sites (e.g., PM$_{10}$ ∼70 µg m$^{-3}$ in Antey, Sect. 4.2.3). Another peculiarity of this case study is the fact that, owing to a temperature inversion close to

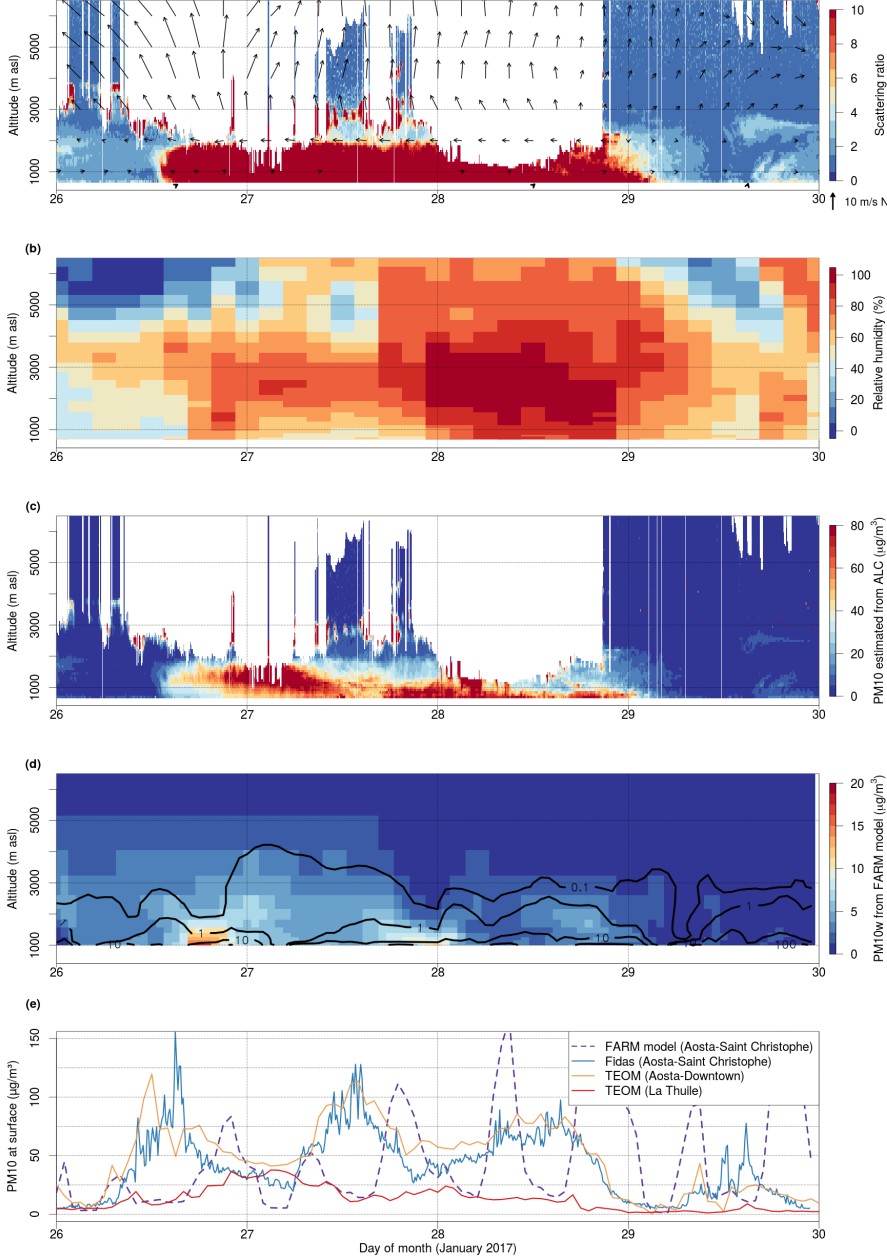

**Figure 10.** Case study of 26–29 January 2017. (a) Coloured background: vertical profile of scattering ratio from ALC in Aosta–Saint Christophe. Arrows: horizontal velocity of the wind measured at the surface and simulated by COSMO at several elevations; (b) Vertical profile of relative humidity forecasted by COSMO; (c) Vertical profile of $PM_{10}$ mass concentration derived from the ALC using the functional relationships; (d) Mass concentration ($PM_{10w}$) from FARM; (e) Hourly and sub-hourly $PM_{10}$ (dry) surface concentration from FARM simulations and observations in Aosta–Saint Christophe, Aosta–Downtown and La Thuile (the y-scale of this panel is extended compared to Fig. 4 and 13).





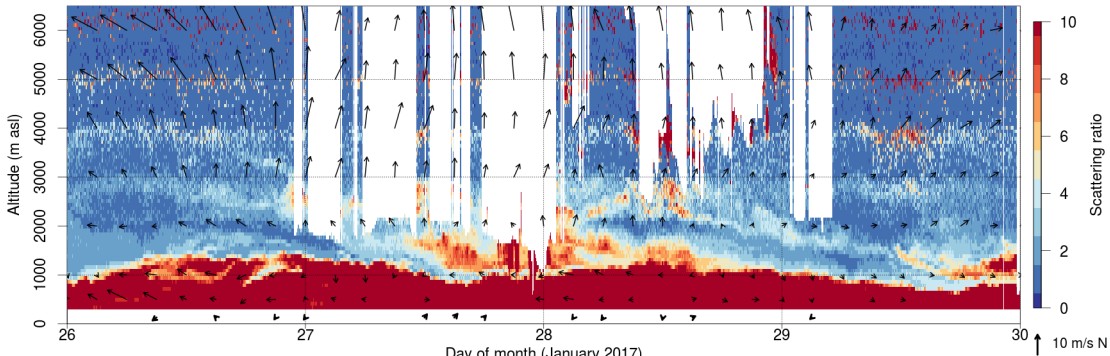

**Figure 11.** Vertical profile of scattering ratio from the ALC in Milan (26–29 January 2017). Arrows: horizontal velocity of the wind measured at the surface (bold, lower arrows) and simulated by COSMO at several elevations (thin arrows).

the ground, the phenomenon could be mainly identified and fully understood by using profiling instruments, such as the ALC, rather than the data from the air quality and weather surface networks. No sun photometric measurements were available for this period due to clouds and major maintenance to the POM-02 instrument.

### 4.2.1 ALC observations

The SR profiles from the ALC in Aosta–Saint Christophe for this winter case are depicted in Fig. 10a and show the sudden appearance of a thick aerosol layer in the afternoon of 26 January. Unlike the previous case, the ALC measurements do not reveal distinct features for each day of the sequence, but rather a continuous and persisting layer during the whole episode. The SR reaches values above 30 in the night between 26 and 27 at altitude, and, more close to the surface, between the evening of 27 and the morning of 29 January. The layer extends up to 2000 m a.s.l., a clear signature of the non-local origin of the air mass

in view of the presence of a temperature inversion close to the ground. Some clouds are visible above and within the aerosol layer, thus further inhibiting the mixing in the PBL. The episode ends on 29 January as quickly as it began, with clearer air taking the place of the polluted air mass starting from above and subsequently eroding the temperature inversion at the surface.

Simultaneous ALC profiles over Milan are depicted in Fig. 11. As opposed to the Aosta Valley, the aerosol layer does not vanish on 29 January, but remains for some days more, although the winds at altitude change their provenance from the west

on that day. Clouds only form from 27 January, presumably allowing solar radiation to trigger a weak breeze tide in the lowest 2000 m on that day, whilst strong stability favours calm wind in the following days.

### 4.2.2 Meteorological variables and back-trajectories

The wind field over Aosta–Saint Christophe, depicted in Fig. 10a, presents a very different pattern compared to the first case addressed (Fig. 4a). Firstly, calm wind is measured for the whole period at the surface. This is due to a temperature inversion in

the lower atmospheric layers in the main valley, as revealed by the thermometers along the mountain slope (Fig. S5, Sect. S5).





Indeed, at the Saint-Denis station, located above the inversion layer, the wind pattern is more representative of the wider circulation: for example, the average wind speed is about $4 \, \mathrm{m\,s^{-1}}$ on 26 January (Fig. S4e, Sect. S5) and the wind clearly turns from west (morning) to east (afternoon), simultaneously with the appearance of the layer. As a further difference with the first case, the forecasted wind at 1000–2000 m a.s.l. does not show any change in direction typical of the thermal winds. For

example, at 2000 m a.s.l. the circulation is continuous, and vigorous (up to $6 \, \mathrm{m\,s^{-1}}$), from the afternoon of 26 to the beginning of 29 January. Indeed, this winter case study interestingly shows that thermally-driven winds are not the only mechanism, especially in winter, driving the advection of air masses from the Po Valley to the Alps. Rather, the synoptical circulation can push the air masses towards the Alpine valleys, as in this case. In fact, the flow clearly reveals its southern origin at elevations above the mountain crest (e.g., 3000 m a.s.l.), where the wind is not channelled within the main valley. At that altitude, the

wind speed is even greater than $20 \, \mathrm{m\,s^{-1}}$. Finally, on 29 January, the measurements in Saint-Denis (gradual increase of the speed of westerly wind) and COSMO simulations (wind reversal at 1000–2000 m a.s.l) correlate with the disappearance of the layer better than observations at the bottom of the valley (calm wind).

Back-trajectories for 26 January are plotted in Fig. S6 (Sect. S5) and indicate transit over the Po basin starting from the morning, which seems to contradict the fact that the layer arrival over the Aosta Valley is detected by the ALC only since

the afternoon. This can be explained by noting that the mean altitude of the trajectories crossing the Po basin during the morning exceeds 1500 m a.s.l. (not shown), and is thus higher than the aerosol layer observed by the ALC in Milan (Fig. 11). The trajectory altitude tends to decrease in the afternoon to the elevations of the polluted boundary layer, leading to effective aerosol transport to the Aosta Valley (the trajectory residence time in the Po Valley PBL being 30–35 hours before arriving over the observing site).

Together with the appearance of the aerosol layer, an increase in the COSMO RH can be noticed (Fig. 10b). The latter remains higher, above typical wintertime deliquescence values (e.g., DRH=54%, D'Angelo et al., 2016), for the whole duration of the episode and never drops below the crystallisation point (e.g., CRH=47%), which can be partly attributed also to the presence of low clouds forecasted by the NWP model, as actually occurred. The advection is detected more clearly by the increase in specific humidity measured at ground (from less than $2 \, \mathrm{g\,km^{-1}}$ to a maximum of $4 \, \mathrm{g\,kg^{-1}}$ on 28 January, Fig. S4b).

### 4.2.3   Mass concentrations and particle measurements at the surface

The mass concentration retrieved within the layer by the ALC (Fig. 10c) is quite variable (from $30 \, \mathrm{\mu g\,m^{-3}}$ at the edge of the layer to more than $100 \, \mathrm{\mu g\,m^{-3}}$ at the core) and reveals the heterogeneous distribution of the particulate inside the layer. FARM predicts a very different scenario, with three separate increases at the end of 26, 27 and 28 January of non-local origin (coloured background in Fig. 10d, much lower than the concentration retrieved by the ALC) and a clear diurnal cycle close

to the surface of local origin (Fig. 10e). The diurnal cycle in the simulations is characterised by two peaks corresponding to the combined effect of traffic rush hours, residential heating and variation of the mixing layer height. Hourly and sub-hourly $PM_{10}$ surface concentration measurements at both Aosta–Downtown and Aosta–Saint Christophe, however, only exhibit one peak at midday. The differences between the model and the measurements at the surface are due to an underestimation of the residential heating (switched on all day during these very cold days) and an overestimation of the traffic road contribution,





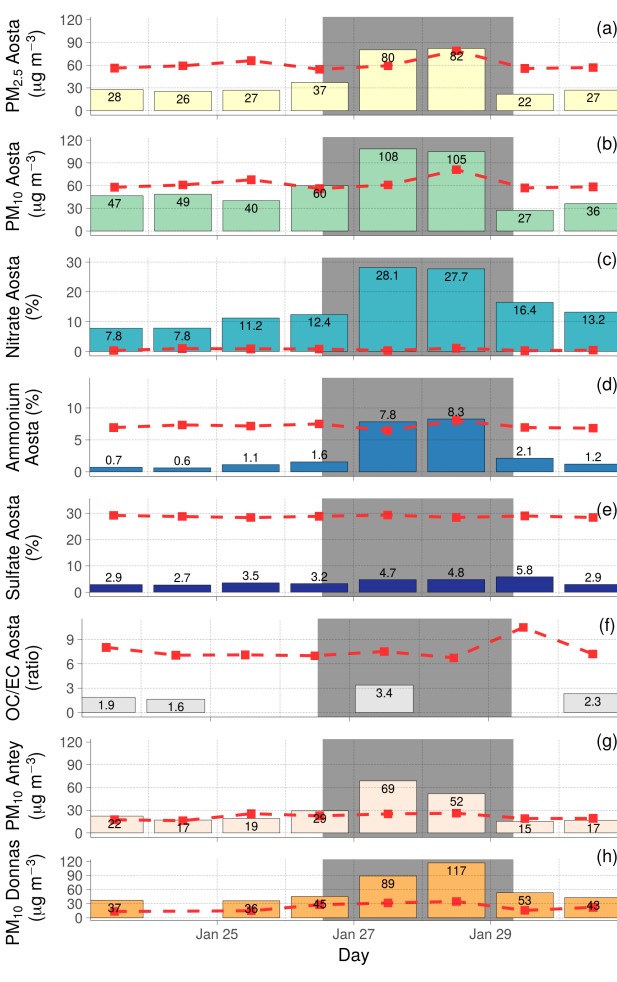

**Figure 12.** Measured (coloured bars) and simulated (dotted line) $PM_{2.5}$ and $PM_{10}$ daily concentrations at several sites of the Aosta Valley (a,b,g,h); percentage concentrations of nitrate (c), ammonium (d) and sulfate (e), and OC/EC ratio (f) at Aosta–Downtown during case study 2 (January 2017). The period when the ALC detects a thick layer above Aosta–Saint Christophe is highlighted with a grey background.



together with an overestimation of the mixing layer height growth at midday by the NWP model. Anyway, Fig. 12 shows that the daily averages of PM concentrations measured at several sites of the region are higher on 27–28 January than on the neighbouring days. Specifically, the increase is similar (more than $40\,\mu\mathrm{g\,m^{-3}}$) for both $PM_{2.5}$ and $PM_{10}$ in Aosta–Downtown, which results from the fact that the increment is mainly driven by particles with diameter less than 2.5 μm. The maximum

$PM_{10}$ concentration ($117\,\mu\mathrm{g\,m^{-3}}$) was measured on 28 January in Donnas (Fig. 12h), which is the closest station to the Po basin. The spatial pattern of the observed increase, not fully captured by the model, is evident in Fig. S7, Sect. S5 and represents a further indication of the Po Valley being the source of the polluted air masses. Moreover, Fig. 12(c,d) shows this increase to be associated to enhancement in Aosta–Downtown of the nitrate and ammonium components (see next paragraph), two key species of the Po Valley secondary aerosol. Finally, while the daily PM concentrations from FARM are comparable, on

average, to the measurements, the modulation of the PM concentration by the advection (peaks) is not captured by the model, whose output is rather constant. Most interestingly, data collected at remote and usually pristine sites also show a remarkable increase: at La Thuile ($PM_{10}$ winter average $7\,\mu\mathrm{g\,m^{-3}}$), the hourly $PM_{10}$ concentration (Fig. 10e) reaches nearly $40\,\mu\mathrm{g\,m^{-3}}$ (some hours later than the appearance of the aerosol layer in Aosta–Saint Christophe) and correlates well with the increasing $NO_2$ concentration (from about $2\,\mu\mathrm{g\,m^{-3}}$ before and after the event to $44\,\mu\mathrm{g\,m^{-3}}$ during the event on a hourly basis) measured

by a co-located detector. Additionally, the mobile laboratory in Antey (winter average $20\,\mu\mathrm{g\,m^{-3}}$) measures increasing daily $PM_{10}$ concentrations with a maximum of $69\,\mu\mathrm{g\,m^{-3}}$ on 27 January (Fig. 12g) and increasing $NO_2$ concentrations from about $30\,\mu\mathrm{g\,m^{-3}}$ to $56\,\mu\mathrm{g\,m^{-3}}$.

For this selected sequence of days, the data collected by the OPC in Aosta–Saint Christophe are additionally available. The instrument reveals a notable increase in the number concentration for particles smaller than 0.5 μm (Fig. S4c) in coincidence

to the arrival of the aerosol layer. The total number concentration (Fig. S4d) gradually increases from few hundres particles $\mathrm{cm^{-3}}$ to 3000 particles $\mathrm{cm^{-3}}$ and decreases again on 29 January.

### 4.2.4   Chemical analyses

Some results of anion/cation analyses performed on daily samples collected at Aosta–Downtown are also reported in Fig. 12 and presented in terms of relative concentrations (ratio between ions mass and $PM_{10}$). As anticipated, the fractions of nitrate

and ammonium drastically increase during the event, reaching values more than double (nitrate) or even eight to ten times as much (ammonium) compared to the concentrations in the days adjacent to the case study. Indeed, wintertime low temperature and high humidity represent the best conditions leading to the formation of ammonium nitrate (Schaap et al., 2004). Besides, this nitrate increase enhances the observation of a lowering of DRH (D'Angelo et al., 2016) that may influence the ALC backscatter. Sulfate also increases, but not as much as nitrate and ammonium, since unfavourable conditions are met during

winter (Carbone et al., 2010). Only one sample was analysed for EC and OC during the event, and the OC/EC ratio increases only marginally, likely due to sample overloading. In general, variations of the aerosol composition are noticeable on 27–28 January and in line with transport from the Po basin. Indeed, high presence of secondary aerosol in the Po Valley has been documented since a long time, most notably nitrate compounds (Schaap et al., 2004; Putaud et al., 2010; Saarikoski et al., 2012; Aksoyoglu et al., 2017). The latter are probably enhanced by the particular atmospheric conditions during the examined





period (Bacco et al., 2017). All together, nitrate, ammonium and sulfate can explain about 40% of the $PM_{10}$ mass during the episode (as a reference, this fraction represents 15% of the $PM_{10}$ mass for non-advection days of January–February 2017, on average), while organic matter (OM, assuming a typical conversion factor of 1.6 between the measured concentration of OC and the unknown concentration of OM, as in Turpin and Lim (2001) and Curci et al. (2015) for urban sites) and elemental

carbon account for a remaining 30% and 5% fraction, respectively (similar percentages are obtained for non-advection days in January–February 2017). Finally, the relative concentration of the other measured ions, allegedly of local origin (e.g., $Na^+$ and $Cl^-$ from road salting, not shown), does not follow the same pattern as observed in Fig. 12. Figures 12c–e also reveal that FARM is not able to reproduce the experimental chemical speciation: nitrate is strongly underestimated, while ammonium and sulfate are strongly overestimated, and the simulations of the OC/EC ratio do not follow the experimental data. This behaviour

is probably to be ascribed to the fact that the SPECIATE v3.2 chemical characterisation implemented in the emission manager is not suitable for the considered sources and/or that the sources, and therefore their chemical profiles, are not accurately identified.

### 4.3 Case study 3: Spring (25–30 May 2017)

This third case, occurring in Spring, is similar to the first one (Sect. 4.1), but is included to represent a third season and

because a more extended observational dataset was available. From a meteorological point of view, a wide high-pressure ridge extended from the Mediterranean Sea to western and central Europe, thus favouring sunny days with afternoon instabilities and thermally-driven winds from the Po basin to the Aosta Valley. At the end of the period, a weakening of the high-pressure area led to increased instability. The overall sequence (25 May – 3 June 2017), only partially described in the following paragraphs (25–30 May), lasted for 10 consecutive days.

#### 4.3.1 ALC observations

Since the establishment of the thermally-driven wind regime, starting from 25 May, a thick aerosol layer is regularly detected by the ALC in the afternoon (Fig. 13a). The layer persists during each night, when the scattering ratio increases up to a value of 20 and clouds systematically form within the layer. This aerosol layer extends from the ground to an altitude increasing from 2.5 km at the beginning of the case study to more than 3 km at the end of the episode. Entrainment of the elevated layer to the

surface in the middle of the day is repeatedly observed by the ALC.

#### 4.3.2 Meteorological variables and back-trajectories

The plain-to-mountain circulation, driving the phenomenon under investigation, is well captured by both measurements at the surface (Fig. 13a, bold arrows) and COSMO forecasts (thin arrows). Eastern winds with speeds $> 10\,\mathrm{m\,s^{-1}}$ are measured in the afternoon till sunset at the surface, while nights are characterised by calm wind. At higher elevations, the wind provenance

turns from the north, at the start of the depicted sequence, to the south.



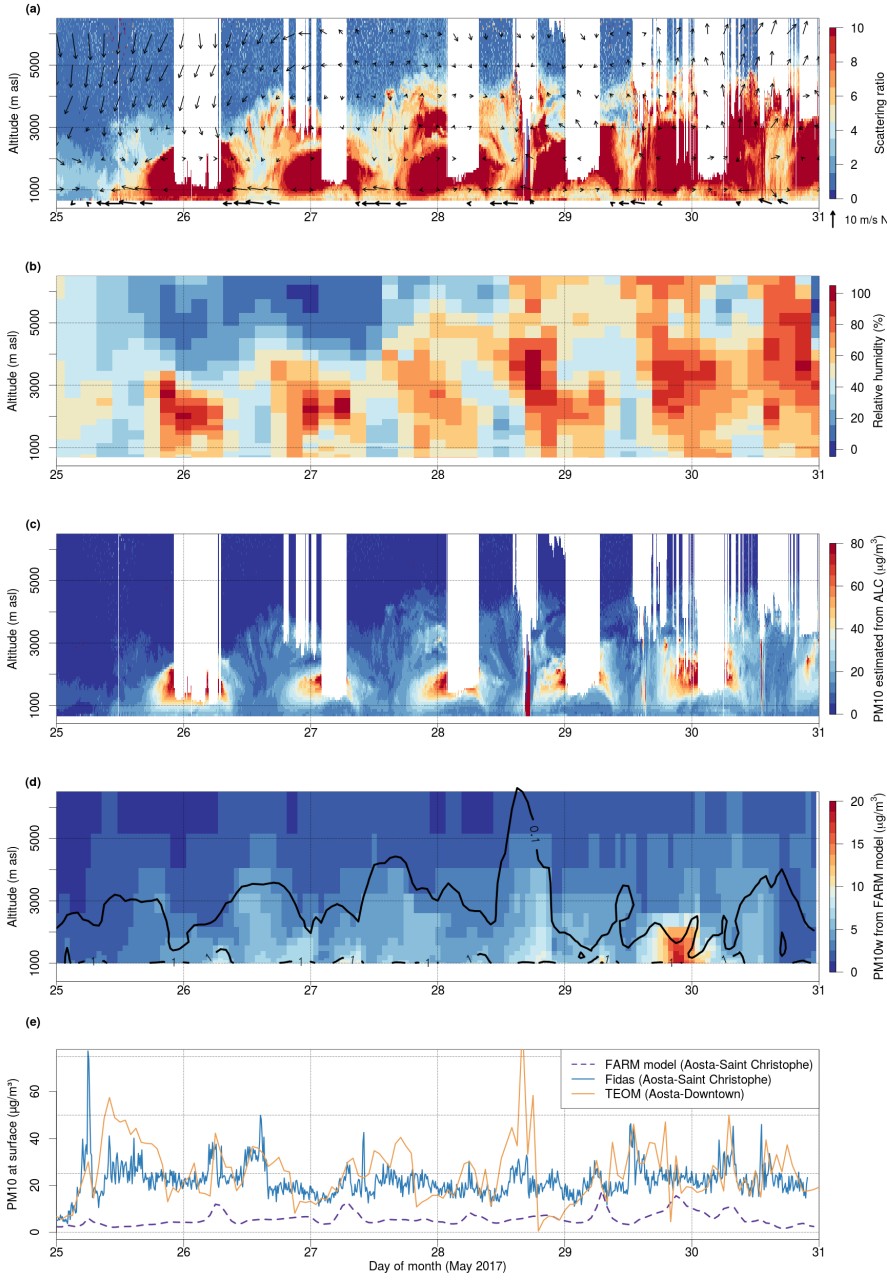

**Figure 13.** Case study of 25–30 May 2017. (a) Coloured background: vertical profile of scattering ratio from ALC in Aosta–Saint Christophe. Arrows: horizontal velocity of the wind measured at the surface and simulated by COSMO at several elevations; (b) Vertical profile of relative humidity forecasted by COSMO; (c) Vertical profile of $PM_{10}$ mass concentration derived from the ALC using the functional relationships; (d) Mass concentration ($PM_{10w}$) from FARM; (e) Hourly and sub-hourly $PM_{10}$ (dry) surface concentration from FARM simulations and observations in Aosta–Saint Christophe and Aosta–Downtown.





The back-trajectories ending over the Aosta Valley on 25 May are plotted in Fig. S9, Sect. S6. The large-scale circulation from the north generally dominates the air mass origin. However, during the day, the low-level thermal circulation becomes strong enough to influence the lowest trajectories, which start to cross the Po Valley in the second part of the day, in line with the simultaneous appearance of an aerosol layer in the ALC measurements. Together with their rotation during this day, the trajectories also decrease their altitude. At the end of the day, the air masses reaching the station have slithered for more than 20 hours on the surface of the Po basin. The analysis of the trajectories for the following days indicates that the air masses keep crossing the Po basin.

As in the first case, COSMO accurately predicts the advection of humid air at the same times as the ALC detects a thickening of the layer (Fig. 13b). At night, the simulated and measured RH exceed 90% at altitude and 80% at ground, respectively (Figs. 13 and S8b). Contrary to RH, specific humidity does not show any daily cycle (Fig. S8b). A sudden SH increase (5 to 10 $g\,kg^{-1}$) is clearly visible on the first day (25 May) at the time of the advection, while the values for the following days are almost constant except on the occasion of short showers (e.g., evening of 28 May).

### 4.3.3 Mass concentrations and particle measurements at the surface

The aerosol mass derived from the ALC is presented in Fig. 13c. The maximum concentration retrieved by this method within the aerosol layer is higher than 60 $\mu g\,m^{-3}$ just before the formation of clouds at night. Again, FARM (Fig. 13d) qualitatively reproduces the afternoon increase of aerosol concentrations owing to transport from the boundaries, however the simulated concentrations are much lower (about 4–5 times) than the retrievals from the ALC and the advection arrival times are anticipated compared to the appearance of the thick layer from the ceilometer.

Hourly and sub-hourly $PM_{10}$ surface concentrations (measured in Aosta–Downtown and Aosta–Saint Christophe and simulated by FARM) are presented in Fig. 13e. FARM correctly reproduces the morning rush-hours peak, but the concentrations are about half those from the PM samplers. The series in Fig. 13e shows an increase of $PM_{10}$ surface concentration during the first day, with persisting high values for the rest of the week, most noticeably during the night. Accordingly, PM daily means in Fig. 14(a,b,g,h) show a distinct increase in the whole region (the concentrations doubles) during the case study compared to the preceding and following days.

The number distribution and total particle number measured by the OPC in Aosta–Saint Christophe are plotted in Fig. S8c and d, respectively. A notable increase in the number concentration during the first day (from less than 200 to more than 800 particles $cm^{-3}$) and in the afternoon of each day, and a decrease in the central part of each day as soon as the valley convection starts and the mixing layer height increases is clearly visible.

### 4.3.4 Chemical analyses

Percentage concentrations of nitrate, ammonium and sulfate are represented in Fig. 14(c–e) and account for about 20–25% of the total $PM_{10}$ mass (as a reference, this fraction represents less than 15% for non-advection days in May–June 2017). Interestingly, relative nitrate concentration does not change much and does not reach the extreme values of the winter case study. Indeed, transfer of ammonium nitrate from particles to gas-phase, which is not measured, is favoured by higher temperatures



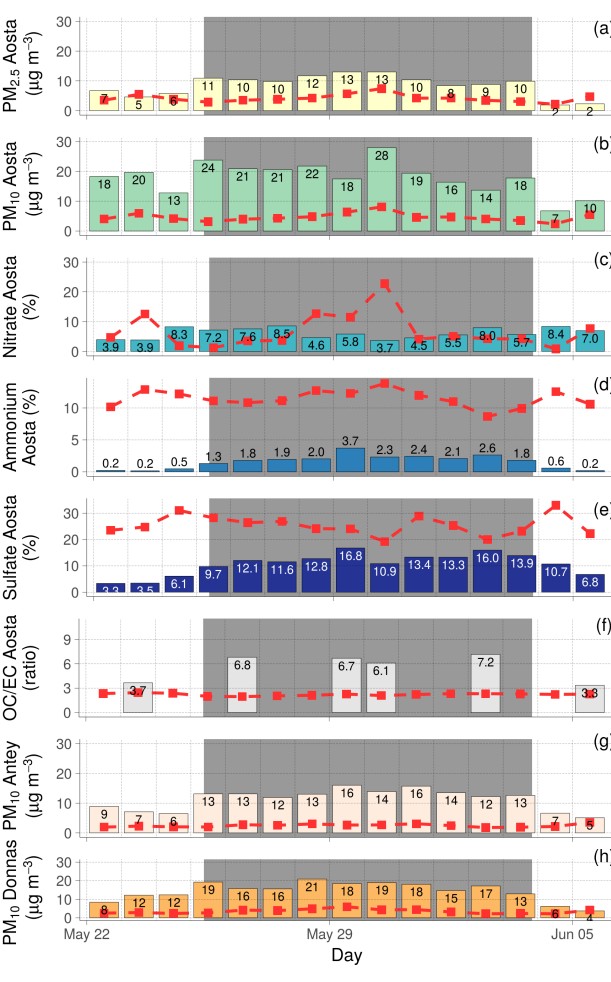

**Figure 14.** Measured (coloured bars) and simulated (dotted line) $PM_{2.5}$ and $PM_{10}$ concentrations at several sites of the Aosta Valley (a,b,g,h); percentage concentrations of nitrate (c), ammonium (d) and sulfate (e), and OC/EC ratio (f) at Aosta–Downtown during case study 3 (May–June 2017). The period when the ALC detects a thick layer above Aosta–Saint Christophe is highlighted with a grey background.





(Saarikoski et al., 2012). Conversely, ammonium and sulfate increase remarkably during the advection, reaching typical concentrations of the Po Valley in that period (Putaud et al., 2010). In particular, the sulfate concentration is much higher and more affected by the advection in May than during the winter case. The role reversal between nitrate and sulfate in case studies 2 and 3 results from the different sensitivity of those compounds to temperature and atmospheric conditions (Carbone et al., 2014).

Again, the contribution of inorganic species from the model does not agree with the analyses: the contribution by ammonium and sulfate is strongly underestimated, while the peaks in the simulated nitrate concentration are not reflected in the analyses.

As for the organic part, OM and EC are the main constituents of the remaining fraction, with about 60% and 6%, respectively. Although the available dataset is rather short, the OC/EC ratio during the event almost doubles (values of 6.1–7.2) compared to the value before (3.7) and after (3.3) the event. This increase of the OC fraction during transport episodes is confirmed by

the long-term analysis (Diémoz et al., 2018).

### 4.3.5   Sun photometer measurements

The same morning-midday-afternoon modulation can be observed in the AOD from both the ALC and the sun photometer (Fig. S8e). The high AOD values in the first and last part of the day (up to 0.30–0.40 at 500 nm) and the decrease in the middle of the day (down to 0.12 at 500 nm) match the appearance of the layer seen by the ALC and its enhancement due to

hygroscopic effects, in accordance with the results of Adam et al. (2012) for a typical site in the Po basin (in that case, for RH=90%, the extinction coefficient increased on average to 180% of the value measured for RH=0%). The steep rise (from 0.8 to 1.7) in the Ångström exponent on the first day (Fig. S8f) and its continuous increase in the following days (up to nearly 2.0) may be attributed to the advection of small particles to the measuring site. The SSA fluctuates around large values (generally between 0.9 and 1.0), typical of weakly light-absorbing or aged aerosol. Most interestingly, the volume distribution (Fig. S8g)

exhibits an abrupt decrease of the peak diameter during the morning hours (e.g., from 0.5 at 6.40 UTC to 0.2 μm at midday on 27 May), strengthening the hypothesis of aerosol hydration at night and dehydration during the day as temperature increases and RH decreases.

### 4.4   Model–measurement discrepancies

Our CTM qualitatively reproduces the aerosol advections in all three case studies. It helps to understand the phenomenon by

allowing switching on/off the non-local sources (boundary conditions), but fails to quantitatively explain the concentrations retrieved by the ALC ($PM_{10w}$) and measured by the air quality network ($PM_{10}$). Notably, the model underestimates the PM mass both in the layer aloft and at the surface, and anticipates (by some hours) the peak concentrations compared to the profiles from the ALC. In particular, the anticipation cannot be due to inaccuracies in modelling the wind speed, since the discrepancies between forecasts and observations are $< 10\%$ during the episodes. A variety of (possibly concurrent) reasons can explain this

behaviour, mainly related to:

1. inaccuracies in retrieving the PM concentration from the ALC backscatter. As mentioned, ALC measures aerosol backscatter, and specific tools were developed and are used here to associate a PM value to it (Dionisi et al., 2018),



but expected error on these estimates are of the order of 30–40%. In addition to the uncertainty related to this ALC-based PM retrieval, other factors also play a role. First, the ALC is sensitive to the total suspended particulate (TSP) of any dimension, whereas FARM simulates the PM concentration for particles with diameter smaller than 10 μm. Secondly, and most importantly, water uptake by aerosol can be prominent, especially when RH>DRH and exceeds the conditions assumed for calculating the functional relationships (RH≤95%). These issues can worsen the comparison between the model ($PM_{10w}$) and the ALC profiles;

2. inaccuracies in the CTM simulations. The emission inventory used within FARM likely underestimates the real emissions, as also reported in other cases, and for different models, in the scientific literature (e.g., EMEP, 2016; Uchino et al., 2017). In particular, the boundary conditions could not be accurate enough for our aims owing to the abrupt change of the national emission inventory grid resolution (12 km) to the local scale (1 km). This issue can affect the comparison between the model (dry $PM_{10}$) and surface measurements. Additionally, the aerosol hygroscopicity may be not optimally simulated by FARM, e.g. due to a wrong characterisation of the chemical properties of the modelled aerosol, which again impacts on the comparison between the vertical profiles from the model and the ALC.

The case studies described here show that several of the previous points most likely play a significant role. Some sensitivity tests were performed addressing point 2. In particular, for the first case study (August 2015, Sect. 4.1) we performed two additional tests. 1) We doubled the PM concentrations from the boundary conditions to assess the sensitivity of the simulated vertical profiles to the accuracy of the national emission inventory and to the transport from outside the administrative boundaries of the Aosta Valley, while leaving the regional emission inventory unchanged. The perturbation of the boundary conditions used for the test may appear excessively large, however this choice could be supported by the fact that the resolution of the national inventory grid is much coarser than the local one, which may be a source of inaccuracies. 2) We employed two different, more empirical, parametrisations of the aerosol hygroscopicity to recalculate water uptake by aerosol. As shown in the Supplementary material (Sect. S7), the results of the two tests support the hypothesis that both the national inventory and the parametrisation of the hygroscopic effects in the model are responsible for the discrepancies between simulations and measurements in the first case study. Doubling the boundary conditions also slightly improves the comparison between simulations and measurements at the surface for the winter case study (first introduced in Sect. 4.2), although some discrepancies in the geographic distribution of the concentrations persist (Sect. S7, Fig. S12), probably due to inaccurate NWP input data (e.g., overestimation on 25 January 2017 and underestimation on the following days).

Finally, it is worth to mention that a small fraction of the detected secondary particulate might form locally due to heterogeneous chemical reactions taking place on the advected particles themselves (e.g., Gilardoni et al., 2016; Kim et al., 2018; Lim et al., 2018). These dynamics could contribute to the observed underestimation, however they are too complex to be simulated by present CTMs. Further efforts on this topic are scheduled for the future.





## 5  Conclusions

We investigated the phenomenology of recurrent episodes of wind-driven arrival of aerosol layers in the northwestern Italian Alps, and specifically in the Aosta Valley. The analysis was performed by combining a multiple-site, multiple-sensor measurement dataset with modelling tools. Through a deep examination of three case studies, specifically-selected within a 3-year dataset as clear examples of the phenomenon under investigation, we can provide the following answers to the scientific questions driving the study (see Introduction).

1. What is the origin of the aerosol layers detected in the northwestern Alps?

   All results agreed in showing these episodes to be associated to the arrival of polluted air masses originating from the Po basin, one of the EU pollution hotspots. To reach this conclusion, we examined wind flows from both the experimental (surface observations of the wind velocity from the meteorological network at multiple elevations) and modelling (high-resolution NWP models, back-trajectories and CTM simulations) perspective. Interestingly, in one case (Sect. 4.2.2), calm wind measurements at the bottom of the valley during a cold-pool episode could give the mistaken impression that the aerosol originated from local sources, since the circulation in the lowermost levels was inactive, while the wind was blowing undisturbed above the temperature inversion. However, the ALC capacities of sounding the vertical profile of the atmosphere, together with the experimental/modelling data at different elevations, turned out to be a substantial benefit for the clear understanding of the phenomenon;

2. What conditions are favourable to the aerosol flow into the valley?

   We show that these advections are due to thermally-driven winds or synoptic flows from the east (Po basin) to the west. These meteorological conditions are frequently met, especially during fair weather days in the warm period of the year. Comprehensive statistics of the cases provided in the companion paper (Diémoz et al., 2018) exploiting the full 3-year record of ALC measurements, show these conditions to occur at least 50% of the days in summer and spring. We expect the frequency of the advections to increase with increasing proximity to the source (Po basin);

3. How do the advected aerosol layers evolve in both altitude and time?

   Thanks to the monitoring capacity (24/7) of the ALCs, we could follow the evolution of the aerosol layer in both altitude and time. We show that the advected aerosol layers can extend up to 4000 m a.s.l. in the warm season, which incidentally points out the potential impacts of aerosol dry and wet depositions on the remote, high-altitude ecosystems. On the other side, the altitude of the layer sounded by the ALC is a clear indication that the emissions are not local. As for the evolution in time, the layers were usually detected to arrive over Aosta in the afternoon, when the plain-mountain thermal regime established. However, the backscatter from the ALC was found to reach its maximum during the night, when water uptake on aerosol took place and clouds could frequently form within the aerosol layer. In the days following the advection episodes, a residual layer was seen to sink towards the surface, with possible, further impacts on the air quality;





4. What is the impact of the transported aerosol on PM surface concentrations and chemical composition?

An important increase in $PM_{10}$ and $PM_{2.5}$ was detectable during the investigated advections, with up to 80 μg m$^{-3}$ of $PM_{10}$ likely transported in Donnas (Fig. 12). The size distribution of the advected particles generally peaks in the accumulation mode, with a diameter of few tenths of μm (as observed by both the OPC, at the surface, and the sun photometer, in the uppermost layers). Moreover, this kind of particulate is weakly light-absorbing (sun photometer). Chemical analyses reveal these layers to produce an increase of the secondary inorganic fraction, composed by nitrate, sulphate and ammonium, i.e. three typical compounds found in the Po Valley atmosphere, and with low deliquescence RH. Weak local formation of secondary particulate could not be excluded during episodes of severe advection (e.g., case study 2), probably also due to aqueous phase chemistry. However, including these latter processes in current CTMs is still challenging. In one of the case studies, the OC/EC ratio was also observed to increase, a possible sign of the transport of organic compounds from the Po basin;

5. Are the presently used chemical transport models able to reproduce and explain the observations along the vertical profile?

In this investigation, the FARM model was very useful to interpret and complement the observations, and particularly to evaluate the relative weight of local and non-local contributions. Notably, FARM could reproduce the observed arrival and persistence of elevated aerosol layers and it correctly attributed them to sources external to the Aosta Valley. However, the timing and absolute values of the advections were poorly reproduced, with anticipations (in time) and underestimations (of aerosol concentrations) compared to the measurements. On the basis of a sensitivity study, these issues were attributed to both water uptake on those highly-hygroscopic particles, not fully taken into account in the model, and deficiencies in the emission inventories, especially owing to the coarse resolution of the national one (12 km). From a modelling perspective, the observation-based results of this work represent a motivation to improve the emission inventories, thus enhancing the reliability of the CTM simulation. In turn, this would allow extending the findings of this work to a wider domain, not covered (or not fully covered) by observations.

The phenomenology described in detail in the current study through the selected case studies has been further investigated in a companion paper (Diémoz et al., 2018), exploiting the complete observational dataset over the period 2015–2017, complemented by a long-term simulation by the FARM model. This wider dataset, inspected by statistical techniques and classification schemes, allowed a quantitative evaluation of the long-term impact of the aerosol transported from the Po basin to the air quality in the northwestern Alps. Still, future work is needed to investigate possible local and basin-wide strategies to effectively mitigate this impact.

*Data availability.* The ALC data are available upon request from the Alice-net (alicenet@isac.cnr.it) and E-PROFILE (http://data.ceda. ac.uk) networks. The sun photometer data can be downloaded from the EuroSkyRad network web site (http://www.euroskyrad.net/index. html) after authentication (credentials may be requested to M. Campanelli, m.campanelli@isac.cnr.it). The measurements from ARPA air




quality surface network are available at the web page http://www.arpa.vda.it/it/aria/la-qualit%C3%A0-dell-aria/stazioni-di-monitoraggio/inquinanti-export-dati. The wind data in Milan refer to the ClimateNetwork®weather station Milan Bicocca and were kindly provided by Fondazione OMD. The weather data from the Aosta–Saint Christophe and Saint-Denis stations can be retrieved from http://cf.regione.vda.it/richiesta_dati.php upon request to Centro Funzionale della Valle d'Aosta. The MAIAC data were made available by Alexei Lyapustin (NASA). The rest of the data can be asked to the corresponding author (h.diemoz@arpa.vda.it).

*Author contributions.* HD, FB and GPG conceived and designed the study and contributed to the interpretation of the results. TM supplied the meteorological observations and numerical weather predictions. GP performed the chemical transport simulations. DD provided the ALC functional relationships and assistance on the Rayleigh ALC calibration with inputs from MH. SP carried out the EC/OC analyses and IKFT helped with the interpretation of the chemical speciation. MC supplied the POM calibration factors. FB and LDC prepared the satellite radiometer data. LDL and LF provided the ALC data from Milan. HD analysed the data and wrote the manuscript with contribution from FB, GPG, and all co-authors.

*Competing interests.* The authors declare that they have no conflict of interest.

*Acknowledgements.* The authors would like to thank: A. Brunier, G. Lupato, P. Proment, S. Vaccari, and M.C. Gibellino for carrying out the chemical analyses; M. Pignet, C. Tarricone, and M. Zublena for providing the data from the air quality surface network; Fondazione OMD and Centro Funzionale della Valle d'Aosta for the additional meteorological series; Alexei Lyapustin for the MAIAC data. The authors would like to acknowledge the valuable contribution of the discussions in the working group meetings organised by COST Action ES1303 (TOPROF). They also gratefully acknowledge the Institute for Atmospheric and Climate Science, ETH Zurich, Switzerland for the provision of the LAGRANTO software used in this publication.





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
