# Peer review of "Transport of Po Valley aerosol pollution to the northwestern Alps. Part 1: phenomenology"

_Atmospheric Chemistry and Physics, 2018_

## Referee Comment (RC1) · Anonymous Referee #2 · 12 Nov 2018

Review of H. Diemoz et al., Transport of Po Valley aerosol pollution to the northwestern Alps. Part 1: phenomenology

**General comment**

The manuscript describes in detail the phenomenon of aerosol transport from the Po basin into the Aosta valley, investigated both by a fairly comprehensive instrumentation and from a modeling perspective. This effect is of universal importance for air quality dynamics in Alpine valleys. While the study does not reveal significant new findings about the phenomenon, it adds another valuable data set and discussion to the scientific literature.

The manuscript is well structured and written coherently, the scientific questions are clearly set at the beginning and the analysis is focused on the their respective answers in the conclusions. Three case studies are investigated thoroughly with respect to available measurements and models.

Besides some very minor comments below, I do not see any further obstacle on the way for publication in ACP.

**Specific comments**

p9, Fig. 4 and all subsequent figures showing heat maps. The blue-yellow-red color maps (diverging color maps) are not ideal for the sequential type data of e.g. backscatter ratios. Sequential color maps with monotonous increase in luminance would be a better alternative here.

p20, l12pp. If only daily averages from August 26 until August 31 are considered, as for the hourly data in Fig. 4, the increase is less pronounced.

p22, Fig. 9b. The red/blue contours are really difficult to distinguish, but I also acknowledge this might be a hard visualization task.

p22, l5pp. The winter study seems a little more complex than the summer/spring studies. As the authors point out, the synoptic wind from the Po basin is mainly above the very stable PBL, so are the Aosta aerosols really all advected and mixed down to the surface? Maybe the contribution of local emissions is of more significant relevance here? Indeed, the daily cycles of measured PM10 surface concentration in Aosta seem to be influenced by local emissions (traffic, heating, etc.). However, I am no expert in atmospheric chemistry to evaluate the significance of the Nitrate and Ammonium percentages during Jan 27 and 28 in Aosta as an indication for air mass origin.

p34, l17. Maybe the two cases for air quality degradation could be distinguished more clearly here, i.e. thermally driven winds in summer/spring and synoptic winds in winter in stable PBL conditions with no surface wind.

**Technical comments**

p16, l11. … in a few hours.

p34, l29. … regime is established.

---

## Referee Comment (RC2) · Anonymous Referee #1 · 13 Nov 2018

General

The manuscript presents three case studies of the exchange of polluted air masses between an Alpine valley and the foreland (Po basin). The case studies are based on multi-site, multi-instrument field data complemented by numerical modelling. They cover three different seasons. The case studies nicely illustrate the complex transport phenomena that occur during the episodes which each last several days. Qualitative agreement is quite good, however, quantitative disagreement between model and observations leads to various hypotheses for the reasons, which require further study and improvements.

The paper is carefully written, well-structured and nicely illustrated. Its content is very relevant in the context of the complex dynamical and chemical (transport) processes in

and near mountainous terrain.

I strongly encourage publication in ACP, pending the few minor corrections below.

Specific comments

The authors extensively use lidar backscatter (scattering ratio) for tracking the aerosol load of the valley atmosphere (Figs. 4a, 5, 10a, 11, 13a). This parameter is well correlated with relative humidity, which is strongly temperature dependent, but less so with absolute humidity SH (Fig. S8b). SH is a better indicator of air mass transport (and particulates) than RH, i.e. I would expect to see more variation in SH when Po basin air arrives to replace the valley air mass. This is visible for example in the afternoon on 28 May. Could you discuss the constancy of SH with the arrival of the Po valley air mass and its implications for the humidity profile in more detail?

Another aspect is the vertical extent of the scattering ratio in the late afternoons of case study 1 and 3. The wind field indicates strong winds in the lower few hundred meters above valley floor (up to ca. 1200 m), while higher up winds are rather weak or calm (at 2000 m). Yet the polluted air mass almost instantaneously reaches from 1000 to 2000 m upon arrival. With a wind shear from 10 m/s to 1 m/s over 1000 m can we expect quasi-simultaneous arrival of polluted air on all altitudes? Is this front-like structure real or is it an effect of radiative cooling which increases RH and particle growth throughout the valley atmosphere? Or – alternatively – is this an artifact of the combination of real backscatter measurements with the modeled wind field?

The back-trajectories are good indicators of the regional origin of the polluted air. The graphical representation of all heights up to 4000 m hides, however, the details of the low-level (<1500 m) transport route of the air masses within the valley. I suggest to show additional afternoon graphs only for these low levels.

The COSMO-I2 model with 2.8 km grid resolution might still be too coarse for a detailed 3-d simulation of the valley atmosphere with its various mixing processes. With the

valley width of 4 km at Aosta, two grid points fit into the valley cross-section at the floor. This may be insufficient for resolving the complex 3-d flow field of an Alpine valley, and may be another explanation of a part of the discrepancy between model and observations.

Technical corrections

P13L7 omit "the" – should read "covering central and southern Europe"

P22L4 Valley-mountain (and sea-land) breezes are... (shift the bracket)

P27L20 hundreds

Fig. 7, caption I could not find the link to the video

Supplement:

P11L15 replace "grow" with "growth"

Fig. S10 mention the PM10 units for all graphs, not only for a) and b). This can be done in the caption.

---

## Referee Comment (RC3) · Anonymous Referee #3 · 21 Nov 2018

The paper analyses 3 cases of transport from the Po Valley to NW Alps with several in-situ (4 sites in Aosta and in pristine environment) and REM instruments from ground and space. All these measurements are extensively documented in the paper and in the supplement, allowing to have a broad view of the pollution events. After having validated the meteorological model COSMO, a chemical transport model (FARM) is evaluated. This paper is a clear documentation of middle range pollution events affecting the alps and the evaluation of model's performances under these conditions is very interesting.

General comments:

- Figures do not always allow to verify the descriptions or conclusions of the study. For example, higher aerosol concentrations are measured during the case studies (Fig.

[Figure]

4e, Fig. 8, Fig. 13e Fig. S8,. . .), but the figures do allow to be sure that the increase in aerosol load is really specific and cannot be attributed to usual/local fluctuations. In other words, it is not possible to see the differences between the period of the case studies and the period without influence from the Po Valley. I do understand that figures similar to Fig. 4 cannot display longer period of time, but another solution should be found! The time period covered by Fig. 12 is fine.

- FARM leads to too low PM10 concentration in summer and in spring, but seems to work better (even if it does not reproduce the diurnal cycle correctly) in winter (case study 2). Is there an explanation for this difference between summer and winter?

- Why FARM regularly has time shift in its estimation (for example in Fig. 13 for case study 3)? Fig. S10 shows that modification of the PM concentration from the boundary conditions and the modification of the hygroscopic growth have no impact on the time of the aerosol increase. Minor comments:

- P. 1 line 14: "and hygroscopic": the begin of the sentence is a comparison, so that the meaning of this last 2 words is not obvious.

- P. 1-2, lines 22-1: for clarity purpose I would add a TO: "likely owing to deficiencies in the emission inventory and TO particle water uptake not fully taken into account"

- Figure 4d + p. 19 lines 2-7: 1) "PM concentration from non-local sources is represented by the coloured background and the effect of local sources by the contour line, at logarithmic steps;" The logarithmic steps are hardly visible on Fig. 4d (I had to discover that they exist on Fig. 10 to find them on Fig. 4). Could you perhaps use a color scheme with a legend ? Without this, it is not evident that the non-local sources exceed the local ones.

- P. 16 line11: delete the "," after "scattering ratio" + give some indication on the altitude of the described layer.

- P. 16 lines 12-19: I do not really understand your description: from the end of 26th of

September, the lowest layer measured by ALC already shows a SR of about 6, so that the "lower levels, with potential consequences on the surface air quality" are already impacted before sunrise on the 27th of September. Your description does not really describes why the SR decrease in the middle of the days. Is the aerosol rich layer entering the ML and contributes then to higher aerosol load in this ML (not observed in Fig 4a), is it dispersed to higher altitude due to the thermal convection or dispersed horizontally ? Please clarify.

- P. 19: is it possible that the observed shift (from some hours to $\frac{1}{4}$ of a day) of the maximum concentration between FARM and ALC can be due to an overestimation of the local effects in comparison to the non-local ones ?

- Figure 8 and p. 6 lines 18-20 + p. 20 lines 13-15: the measured PM2.5 and PM10 during the case study are similar to the annual mean concentration (p.6). An increase of PM2.5 and PM10 is clearly visible from the 26th of August to the beginning of September (at least at Aosta), but the readers cannot be sure that this does not correspond to a usual fluctuations of the aerosol load. Similarly, it is not really visible that FARM predicts an increase of the PM10 concentration at Aosta during the case study.

- Fig. 4e + page 20 lines 8-9: : the PM10 daily cycle does not correspond to the Po Valley air masses transport seen by the ALC. The given explanation relates to the mass losses occurring in TEOM due to secondary aerosol volatility. Is it expected that this volatility should be different between nighttime and daytime (i.e. larger during the night) in order to explain the different diurnal cycles ?

- Fig S5 is cited before Fig. S4.

- P. 25 line 2: it has however to be mentioned (and perhaps explained) that the wind direction at Saint-Denis (Fig. S4e), after turning to the east during the afternoon of the 26th of January, turns again to west on the 27th in the morning and stay globally at west during the rest of the case study.

- P. 25 lines 10-12:the increase of the wind speed at Saint-Denis on the 29th is not very large (about 2 m/s on the 26th in the morning, increase to 4 m/s on the 26th in the afternoon, about 1 m/s until the end of the 28th, then 2 m/s), so that the correlation with the disappearance of the layer is not really explicit.

- Fig. S6 + p. 25 lines 13-15: the described transition (for the lowest levels I suppose) is quite difficult to see: too much backtrajectories, the lowest ones being represented under the highest ones. A figure allowing to see not only the trajectories but also the change of the altitude over the Po Valley (lines 15-29) would perhaps be more interesting)

- Fig. S4d: the particle number concentration increase during the case study is clearly visible. It is however not possible to determine if this is a peculiar or a normal event occurring regularly during winter.

- Fig. S9 is cited before fig. S8.

- P. 32-33, lines 31-6:The hygroscopic growth of the aerosol can clearly be a cause of the discrepancy between FARM and ALC, even if FARM is also not able to reproduce PM measurements and their diurnal cycles. The difference of the measured size fraction (TSP for ALC and PM10 for FARM does however not seem to be really relevant since: 1) Aerosol from the Po Valley are described as small ones in the paper, see for ex. Case study 2, where the increase in PM10 is similar to the one of PM2.5 (Fig. 12), 2) the size distribution measured by the OPC and sun photometer peak in the accumulation mode and do not show big particles,

- P. 33 lines 7:13: How good is the estimation of the PBL height in the CTM simulations ? Could a bad estimation of the PBL height be also a source of inaccuracies ?

- P. 34 line30-32: as explained previously, I do not see any evidence of a residual layer sinking towards the surface in this study.

---

## Referee Comment (RC4) · Anonymous Referee #4 · 29 Nov 2018

General comments:

The paper analyzes the impact of pollutants transport from the Po Valley to the air quality on the Alpine region. Three selected cases studies of transport over the Aosta Valley are selected among a 3 years period and analyzed in detail by means of a wide-ranging set of observations and numerical simulations. Moreover, the observational data are used to evaluate the performances of the FARM chemical transport model. These results provide an interesting and comprehensive description of the complex phenomena of mountain-valley exchange and deserve therefore to be published. Even if the paper is well written, though, the large amount of data presented in both the main paper and the support material makes the manuscript dispersive and fragmented.

I would suggest to further select the figures that are useful to support their message

and eliminate redundancies. For example, I invite the authors to carefully evaluate if the section 4.1.5 is bringing any relevant information: the analysis seems not fully convincing, is limited to a single case and the information that intends to bring is already conveyed by the FARM model. Similarly, some figures can be just substituted by a short mention in the text (I would reconsider the relevance of fig. 3, S2c, S4e, S5, S12).

Specific comments:

Page 2, line 19-22: Basing on the results of the FARM comparison I would rather highlight that the paper aimed at an evaluation of the model and then mentioning that, despite its limitations, it still brought insights on the cases, supporting the observations. Similarly, I would rephrase the paragraph in the conclusions, accordingly (Page 35, line 14 onward)

Page 4, line.3: Do you really mean that is the partition between local and non-local sources that help to assess the impact of air pollutants on health, climate and ecosystem?

Page 14, line 21. “Therefore, to reduce errors in trajectories with increasing running time, we limit the computation to this duration”. The sentence between commas, in this context, is misleading. Rewrite as “Therefore, we limit the computation to this duration”

Page 18, Figure 6. It would be more meaningful to show the trajectories at 18:00 UTC for this case, since at this time you observe the start of the intense layer arrival and the trajectories are also showing a larger impact from the Po Valley.

Page 16, line 14: Mention here that you see these surface impacts in your measurements (Fig.8 ). Also, what will happen if you average the hourly measurements of TEOM on a daily interval? Will it be consistent with the results of figure 8? Please note that the time interval shown in figure 4 seems to be, compared to figure 8, in the middle of the event (Starting from 26 August and ending at the end of 3 September), so that is not possible to identify any difference with the previous or following phase.

Page 17, line 33: Do you mean here that the transport from the Po Valley persists at every hour along the following days? Indeed, considering the evolution of the layer as seen from the measurements, it would be useful to show which is the typical transport during the whole event at different hours, for example adding the average position of the trajectories (or the PDF of the position in case of large variability) on the considered period. If the authors want to still keep the trajectory of the first day of the episode as a reference for the layer arrival, I would suggest, to avoid burden the paper with additional plots, showing just the synoptic hours instead of a 3-hour time step; it should still be enough to demonstrate the transport evolution. This would also allow comparing the first and third transport type with the mean transport behaviour of the second case study, when the layer persists for the whole duration of the event.

Page 21, line 15: Can you add some reference for the typical Angstrom exponent for the Aosta valley in comparison?

Page 25, line 18 (but same for page 30, line 6): How do you estimate this residence time, is it an average of the time spent by each trajectory over the Po Valley at all levels? From the shown figure (even if it is difficult to distinguish), the transport, including the part outside the PBL, seems faster (around 20 hours or even less). Can you be more specific?

Page 25, line 33-35: How do you justify these affirmations?

Page 32, line 29: It would be interesting to see this comparison indeed since, from figure S1, it seems that the model has the tendency to see easterly winds more often and with higher intensity respect to the observations. Is this comparison limited just to the surface measurements? May it be that there are problems in the higher layers (where most of the pollution layer transport is coming from)? For example, in the case study 2, when the winds were weaker both at the ground and at higher layers, the time evolution of the FARM simulation of Figure 10d was in better agreement with the observations. Is also interesting to note that the time evolution of the vertical distribution

of PM10 from FARM is in better agreement (especially the local contribution) with the surface TEOM measurements rather than with the elevated layers observations.

Technical corrections:

Page 4, line 10: “will be quantified in a companion. . .”

Page 4, line 28: For easier reading, specify which valley are you talking about. In the same paragraph, you are referring to both the Po valley and the Alpine valleys.

Figure 2: The dashed lines are not needed.

Page 9, line 21: “. . .altitude of the extinction coefficient.”

Page 21, line 22: Refer also to the figure S8.

Page 34, line 6: Add a “:” after “(see Introduction)”

Page 35, line 19: “. . .these issues may be partly attributed to. . .”

---

## Author Response (AR2)

**Response to Anonymous Referee #1**

**General Comment. The manuscript presents three case studies of the exchange of polluted air masses between an Alpine valley and the foreland (Po basin). The case studies are based on multi-site, multi-instrument field data complemented by numerical modelling. They cover three different seasons. The case studies nicely illustrate the complex transport phenomena that occur during the episodes which each last several days. Qualitative agreement is quite good, however, quantitative disagreement between model and observations leads to various hypotheses for the reasons, which require further study and improvements. The paper is carefully written, well-structured and nicely illustrated. Its content is very relevant in the context of the complex dynamical and chemical (transport) processes in and near mountainous terrain. I strongly encourage publication in ACP, pending the few minor corrections below.**

We thank the reviewer for taking the time to revise our manuscript and for his/her pertinent comments. Our reply to these is given hereafter (the text in italics represents a citation of the revised manuscript and the figure references follow the updated numbering).

**Referee's comment 1. The authors extensively use lidar backscatter (scattering ratio) for tracking the aerosol load of the valley atmosphere (Figs. 4a, 5, 10a, 11, 13a). This parameter is well correlated with relative humidity, which is strongly temperature dependent, but less so with absolute humidity SH (Fig. S8b). SH is a better indicator of air mass transport (and particulates) than RH, i.e. I would expect to see more variation in SH when Po basin air arrives to replace the valley air mass. This is visible for example in the afternoon on 28 May. Could you discuss the constancy of SH with the arrival of the Po valley air mass and its implications for the humidity profile in more detail?**

Author's response 1. We thank the reviewer for this comment, which gave us the opportunity to study in more depth the measured and the model simulated SH fields, their difference and the relation of this difference to model inaccuracies in reproducing the wind fields. In fact, this further analysis showed that the model-based evolution of SH is strictly linked to the simulated wind regimes (as expected) and further revealed some inaccuracies of the model.

In fact, we found *differences between the simulated and measured daily cycle of specific humidity* (e.g., Fig. S15 for case study 1 as an example). *In particular, the measured SH usually increases during the first advection day as a result of the transport from source areas with more stagnating conditions, but stays rather constant for the rest of the episode. Conversely, COSMO yields larger dynamics, with SH maxima in the late afternoon and subsequent decrease.* This is likely due to the fact, that *COSMO overestimates the nighttime drainage winds (katabatic winds), as noticeable from Fig.S13* (the figure refers again to

case study 1). *This might trigger enhanced cleansing of the lower atmospheric layers during the night as simulated by FARM, but undetected by the ALC.*

[Figure]

**Figure S15:** Comparison among specific humidity measured at Aosta–Saint Christophe and simulated by COSMO at two different altitudes (surface and 2000 m a.s.l.).

[Figure]

**Figure S13:** Zonal component of wind velocity during episode 1 (August 2015) from COSMO (1000 and 2000 m a.s.l.) and two surface stations (Aosta–Saint Christophe and Saint-Denis). Positive $U$ represent wind from the west, negative $U$ wind from the east.

Following the reviewer's suggestions, and our related analysis, Sections 4.1.2 and 4.4 were updated accordingly, as well as the conclusions.

As a final remark, please note that the sharp increase of SH in the afternoon of 28 May is due to rainfalls, as visible from the corresponding ALC panels and also mentioned in the text (former line 12, page 30).

**RC2. Another aspect is the vertical extent of the scattering ratio in the late afternoons of case study 1 and 3. The wind field indicates strong winds in the lower few hundred meters above valley floor (up to ca. 1200 m), while higher up winds are rather weak or calm (at 2000 m). Yet the polluted air mass almost instantaneously reaches from 1000 to 2000 m upon arrival. With a wind shear from 10 m/s to 1 m/s over 1000 m can we expect quasi-simultaneous arrival of polluted air on all altitudes? Is this**

**front-like structure real or is it an effect of radiative cooling which increases RH and particle growth throughout the valley atmosphere? Or – alternatively – is this an artifact of the combination of real backscatter measurements with the modeled wind field?**

AR2. We think that the impression of a "front-like" structure mentioned by the reviewer is partly due to the temporal extent of former Figs. 4 and 13, and the resulting "squeezing" of the daily ALC profiles. In fact, if we consider only one day at a time, e.g. 25 May 2017 (case 3, former Fig. 3 in the discussion paper, also reported here below), or 27 August 2015 (case 1, Fig. A here below), we observe a different, and more accurate picture of the phenomenon. These figures, also representative of a general behaviour noticeable during the advection events, show a more gradual growth of the layer characterised by large SR values. In particular, two distinct regions can be identified: a first one, where an initial increase of SR is visible (region 1, light-blue/green colour), and a second one, where the backscatter is rapidly enhanced (region 2, yellow colour). Based on the discussion in Sects. 4.4 and S7, we ascribe these regions to two different physical processes. Region 1 defines the spatio-temporal domain where advection of dry aerosol during the afternoon occurs. Real transport of aerosol is indeed confirmed at the surface, e.g., by an increase of the number of particles detected by the OPC during event days and an increase of mass concentration from the PM monitors, which operate in dry conditions and are thus not influenced by hygroscopic processes. This region (e.g., quantitatively defined by the SR=3 envelope) can be effectively fitted by a smooth curve as a function of time, as done using a sigmoid function in the companion paper (Diémoz et al., 2019, to be submitted to ACP). Conversely, region 2 refers to the domain where hygroscopic growth takes place during the evening/night, as soon as the sun sets and RH increases. Further investigations and parametrisations of this last phase would require the measurement of profiles of different meteorological variables as well as knowledge of the aerosol properties along the vertical, and will be the topic of future studies.

[Figure]

**Figure 3 (discussion paper):** ALC profile on 25 May 2017.

[Figure]

**Figure A:** ALC profile on 27 August 2015.

A second factor mentioned in the reviewer's comment is the wind profile. We think that some artifacts in the modelled wind fields by COSMO are triggered by *the smoothed valley orography used in the NWP model compared to the real one. This is displayed in Fig. S14, showing the difference of the Digital Elevation Model (DEM) used within COSMO and a more realistic DEM (10 m resolution): both valleys and mountain crests are clearly smoothed out by COSMO, with absolute differences well > 500 m (and up to 1000 m)* (Sect. 4.4 of the revised text). *This difference could [...] explain why the altitude of the entrainment zone (i.e., the boundary between the free atmosphere and the boundary layer where the thermally-driven circulation develops) is underestimated by COSMO compared to the height of the aerosol layer detected by the ALC.*

[Figure]

**Figure S14:** Difference between the (smoothed) Digital Elevation Model (DEM) used by COSMO-I2 (2.8 km resolution) and a higher-resolution DEM ("real topography", 10 m resolution).

**RC3. The back-trajectories are good indicators of the regional origin of the polluted air. The graphical representation of all heights up to 4000 m hides, however, the details of the low-level (< 1500 m) transport route of the air masses within the valley. I suggest to show additional afternoon graphs only for these low levels.**

AR3. We thank the reviewer for pointing this out. Following this and the other reviewer's comments, the back-trajectories figures were modified in this way: we plotted in separate panels trajectories ending at altitudes < 2000 and > 2500 m a.s.l. over Aosta–Saint Christophe. Also, following a remark from referee #4 (RC7) to further simplify the figures, we only show back-trajectories for specific times corresponding to the most significant variations of circulation patterns. Finally, according to referee #3 (RC15), for each time selected we added a bottom panel with the trajectory altitude along their journey. An example of the new plots now included in the paper is provided in Fig. 5.

[Figure]

**Figure 5:** 48-hours back-trajectories ending at Aosta–Saint Christophe on 26 August 2015 at 18 UTC at altitudes lower than 2000 m a.s.l. (a) and higher than 2500 m a.s.l. (b). The trajectories are cut at the border of the COSMO model. The colour scale represents the back-trajectory arrival height. Corresponding altitudes of the backtrajectories vs time are reported in the bottom panels. The dots along each trajectory mark a 1-hour step and the black star indicates the trajectory arrival point (Aosta–Saint Christophe).

**RC4. The COSMO-I2 model with 2.8 km grid resolution might still be too coarse for a detailed 3-d simulation of the valley atmosphere with its various mixing processes. With the valley width of 4 km at Aosta, two grid points fit into the valley cross-section at the floor. This may be insufficient for resolving the complex 3-d flow field of an Alpine valley, and may be another explanation of a part of the discrepancy between model and observations.**

AR4. The reviewer raised a highly topical issue. We are aware of this problem and decided to integrate Sect. 4.4 with the following text:

*As a final remark, we also mention that the 2.8 km grid resolution of the COSMO-I2 model might still be insufficient for resolving the complex 3-D flow field of an Alpine valley and is too coarse to reproduce the mountain atmosphere with its various mixing processes. Follow-up studies using next generation NWP models with increased resolution (1 km or lower) would be of great interest. On the other hand it should also be noticed that decreasing the grid spacing below the scale for which turbulence parametrisations have been developed, i.e. modelling the "grey zone" (or "terra incognita", e.g., Wyngaard, 2004), does not necessary lead to better performances. In this context, comparison of high-resolution simulations with our vertically-resolved dataset could represent a challenging future benchmark for this relevant topic of ongoing research.*

**RC5. P13L7 omit "the" – should read "covering central and southern Europe"**

AR5. Done.

**RC6. P22L4 Valley-mountain (and sea-land) breezes are ... (shift the bracket)**

AR6. Done.

**RC7. P27L20 hundreds**

AR7. Done.

**RC8. Fig. 7, caption I could not find the link to the video**

AR8. We added the following link to the caption: `https://doi.org/10.5446/38391`.

**RC9. Supplement: P11L15 replace "grow" with "growth"**

AR9. Done.

**RC10. Fig. S10 mention the PM10 units for all graphs, not only for a) and b). This can be done in the caption.**

AR10. Done.

**References**

Diémoz, H., Gobbi, G. P., Magri, T., Pession, G., Pittavino, S., Tombolato, I. K. F., Campanelli, M., and Barnaba, F.: Transport of Po Valley aerosol pollution to the northwestern Alps. Part 2: long-term impact on air quality, submitted to Atmos. Chem. Phys., 2019.

Wyngaard, J. C.: Toward Numerical Modeling in the "Terra Incognita", J. Atmos. Sci., 61, 1816–1826, doi:

10.1175/1520-0469(2004)061<1816:TNMITT>2.0.CO;2, 2004.

**Response to Anonymous Referee #2**

**General comment. The manuscript describes in detail the phenomenon of aerosol transport from the Po basin into the Aosta valley, investigated both by a fairly comprehensive instrumentation and from a modeling perspective. This effect is of universal importance for air quality dynamics in Alpine valleys. While the study does not reveal significant new findings about the phenomenon, it adds another valuable data set and discussion to the scientific literature. The manuscript is well structured and written coherently, the scientific questions are clearly set at the beginning and the analysis is focused on the their respective answers in the conclusions. Three case studies are investigated thoroughly with respect to available measurements and models. Besides some very minor comments below, I do not see any further obstacle on the way for publication in ACP.**

We thank the reviewer for taking the time to revise our manuscript and for his/her pertinent comments. Our reply to these is given hereafter (the text in italics represents a citation of the revised manuscript and the figure references follow the updated numbering).

**Referee's comment 1. p9, Fig. 4 and all subsequent figures showing heat maps. The blue-yellow-red color maps (diverging color maps) are not ideal for the sequential type data of e.g. backscatter ratios. Sequential color maps with monotonous increase in luminance would be a better alternative here.**

Author's response 1. Following the reviewer suggestion, we now use a more appropriate colour map starting from Fig. 3 (new numbering) and for all subsequent figures. The new colormap (which is similar to the "Parula" one used in Matlab) is sequential, perceptually uniform, colorblind safe and print/photocopy safe.

**RC2. p20, l12pp. If only daily averages from August 26 until August 31 are considered, as for the hourly data in Fig. 4, the increase is less pronounced.**

AR2. The reviewer is right. Following this comment, and similar remarks from referee #3 (RC1 and 10) and referee #4 (RC6), we expanded (and homogenised) former panels 4, 10 and 13 (old numbering) to include the same number of days. At the same time, we also paid attention at introducing into the sequence a "clear" day in order to better show the effect of the advections. We thus opted to show one week of measurements in each of these figures, as a compromise between completeness and detail (e.g.,

of the wind velocity field). The new plots extend from 25 to 31 August 2015 (case 1), from 24 to 30 January 2017 (case 2) and from 24 to 30 May 2017 (case 3), respectively. This allows to better appreciate the difference between event- (clear) and non-event (polluted) days.

Finally, for each episode, the information on the respective seasonal average concentrations were added to the text to provide reference values. Also note that a rigorous assessment of the long-term impact of the phenomenon presented in this part 1 is indeed the purpose of our companion paper (Diémoz et al., 2019) based on a statistical analysis of the complete dataset (2015–2017).

**RC3. p22, Fig. 9b. The red/blue contours are really difficult to distinguish, but I also acknowledge this might be a hard visualization task.**

AR3. Thank you for your suggestion. We made the contour lines thicker now and the revised figure looks better (Fig. 8).

[Figure]

(a)                                                        (b)

**Figure 8:** (a) Average difference between AOD estimated from Aqua and Terra satellites during days 27–31 August 2015 using the MAIAC algorithm. (b) Horizontal wind velocity from COSMO (arrows), vertical velocity (red/blue contours, $\pm 0.1\,\mathrm{m\,s^{-1}}$) over the same domain and the same hours as in Fig. (a).

**RC4. p22, l5pp. The winter study seems a little more complex than the summer/spring studies. As the authors point out, the synoptic wind from the Po basin is mainly above the very stable PBL, so are the Aosta aerosols really all advected and mixed down to the surface?**

AR4. First of all, we updated the introduction of this case study to highlight that indeed the winter episode is more complex than the other ones:

*Although local emissions (e.g., residential heating and traffic, additionally worsened by the temperature inversion), might have also increased in this period, the influence of pollution transport from the Po basin is unambiguous. As a result of the advection, the PM concentrations measured in the Aosta Valley were found to be significant in the whole region (e.g., $PM_{10} > 100\ \mu g\,m^{-3}$ in Aosta–Downtown and Donnas), even at some remote measuring sites (e.g., $PM_{10} \sim 70\ \mu g\,m^{-3}$ in Antey, Sect. 4.2.3, and remarkably higher*

*than the average concentrations in the same period (e.g., 33 $\mu g\,m^{-3}$ for $PM_{10}$ and 23 $\mu g\,m^{-3}$ in Aosta–Downtown in 2015–2017).*

Then, to further support our analysis and data interpretation, we also changed former Fig. S5 in the Supplement. The new figure shows the vertical gradient of *pseudo-equivalent potential temperatures (e.g., Freney et al., 2011) at different altitudes [...], thus providing a rough indication of the vertical extent of the mixed layer. For this second episode, the arrival of a different air mass is revealed by the temperature/humidity sensors along the mountain slope. Pseudo-equivalent potential temperatures at different altitudes are shown in Fig. S5. As clearly noticeable, the spread among the series recorded at 550 m a.s.l. and the ones at higher altitudes remarkably, and quickly, decreases on 26 January, especially during the night, suggesting that the strong (and very shallow) temperature inversion weakens and mixing of the upper aerosol layers down to the surface is favoured.*

[Figure]

**Figure S5:** Profile of pseudo-equivalent potential temperature measured along the mountain slope on January 2017. A weakening of the temperature inversion, and a more mixed boundary layer, are clearly detected from 26 to 28 January, i.e. during the advection episode.

**RC5. Maybe the contribution of local emissions is of more significant relevance here? Indeed, the daily cycles of measured PM10 surface concentration in Aosta seem to be influenced by local emissions (traffic, heating, etc.).**

AR5. We agree that the daily cycle of measured $PM_{10}$ may be partly influenced by local emissions during this episode as well as during other advection events. In this respect, in the companion paper, we indeed study the typical daily cycle of PM concentrations in different conditions (e.g., no aerosol layer detected by the ALC, aerosol layer arriving in the afternoon or leaving in the morning), based on statistical analyses of the long-term dataset. The PM daily cycles in these cases are represented in Fig. 10 (from companion paper). Although the overall daily tendency may vary as a function of the day type (e.g., increasing trend in case of a layer arriving in the afternoon, decreasing trend in case of a layer leaving in the morning), a common, typical feature of all plots is a double daily peak, resulting from both increased local emissions in the morning and late afternoon, and the evolution of the mixing layer height. The daily cycle of Polycyclic Aromatic Hydrocarbons (PAHs), detected at the same site, is also plotted. In fact, as demonstrated in "part 2" of the study (using Positive Matrix Factorisation), PAHs are related to (and are a good proxy of) the local emissions (e.g., traffic, combustion).

[Figure]

**Figure 10 (companion paper):** (a) Daily $PM_{10}$ cycle sampled by the OPC in Aosta–Saint Christophe during non-event days (class A) and days with arriving and leaving aerosol layers (classes C and D), plotted together with the daily evolution of PAH as a proxy of the local sources (dotted line, right vertical axis, in $ng m^{-3}$). (b) Same as (a) using the TEOM in Aosta–Downtown.

During case study 2 (Fig. A), this typical daily cycle in Aosta–Downtown is visible in non-event days, i.e. Thursday 26 and Sunday 29 January 2017, but is missing during event days, i.e. on Friday 27 and Saturday 28 January, where the correlation between $PM_{10}$ and PAHs is lost. We ascribe this behaviour to the advection of non-local, secondary aerosol from the Po Valley, such as ammonium nitrate. In fact, the chemical analyses show a remarkable increase of this compound, which we demonstrate (in the companion paper) to be a clear marker of the typical aerosol transported from the Po basin. This implies that local emissions are not the main effect modulating the $PM_{10}$ concentrations in the winter case study.

[Figure]

**Figure A:** $PM_{10}$ and Polycyclic Aromatic Hydrocarbons (PAH) concentrations measured in Aosta–Downtown during case study 2.

**RC6. However, I am no expert in atmospheric chemistry to evaluate the significance of the Nitrate and Ammonium percentages during Jan 27 and 28 in Aosta as an indication for air mass origin.**

AR6. This topic will be thoroughly and rigorously discussed in the companion paper using the Positive Matrix Factorisation (PMF) technique coupled with the SR profiles from the ALC. There we demonstrate that nitrates (mostly in winter), sulfates (mostly in summer) and ammonium are indeed good markers of the advections from the Po basin. Also, locally-produced secondary aerosols are expected to be *minor contributors to PM$_{10}$, owing to missing sources of precursors in the Aosta Valley.*

**RC7. p34, l17. Maybe the two cases for air quality degradation could be distinguished more clearly here, i.e. thermally driven winds in summer/spring and synoptic winds in winter in stable PBL conditions with no surface wind.**

AR7. Right. This was mentioned in the revised conclusions:

*We show that these advections are due to thermally-driven winds (especially in the warm period of the year, e.g. case studies 1 and 3) or synoptic flows (mainly in the cold season, e.g. case 2) from the east (Po basin) to the west. A more systematic analysis of the flow regimes and their impacts on transport based on comprehensive statistics are provided in the companion paper (Diémoz et al., 2019) exploiting the full 3-year record of ALC measurements.*

**RC8. p16, l11. ... in a few hours.**

AR8. Done.

**RC9. p34, l29. ... regime is established.**

AR9. Done.

**References**

Diémoz, H., Gobbi, G. P., Magri, T., Pession, G., Pittavino, S., Tombolato, I. K. F., Campanelli, M., and Barnaba, F.: Transport of Po Valley aerosol pollution to the northwestern Alps. Part 2: long-term impact on air quality, submitted to Atmos. Chem. Phys., 2019.

Freney, E. J., Sellegri, K., Canonaco, F., Boulon, J., Hervo, M., Weigel, R., Pichon, J. M., Colomb, A., Prévôt, A. S. H., and Laj, P.: Seasonal variations in aerosol particle composition at the puy-de-Dôme research station in France, Atmos. Chem. Phys., 11, 13 047–13 059, doi:10.5194/acp-11-13047-2011, 2011.

**Response to Anonymous Referee #3**

**The paper analyses 3 cases of transport from the Po Valley to NW Alps with several in-situ (4 sites in Aosta and in pristine environment) and REM instruments from ground and space. All these measurements are extensively documented in the paper and in the supplement, allowing to have a broad view of the pollution events. After having validated the meteorological model COSMO, a chemical transport model (FARM) is evaluated. This paper is a clear documentation of middle range pollution events affecting the Alps and the evaluation of model's performances under these conditions is very interesting.**

We thank the reviewer for taking the time to revise our manuscript and for his/her pertinent comments. Our reply to these is given hereafter (the text in italics represents a citation of the revised manuscript and the figure references follow the updated numbering).

**Referee's comment 1. Figures do not always allow to verify the descriptions or conclusions of the study. For example, higher aerosol concentrations are measured during the case studies (Fig. 4e, Fig. 8, Fig. 13e Fig. S8, ...), but the figures do allow to be sure that the increase in aerosol load is really specific and cannot be attributed to usual/local fluctuations. In other words, it is not possible to see the differences between the period of the case studies and the period without influence from the Po Valley. I do understand that figures similar to Fig. 4 cannot display longer period of time, but another solution should be found! The time period covered by Fig. 12 is fine.**

Author's response 1. Following this comment, and similar remarks from referee #2 (RC2) and referee #4 (RC6), we expanded (and homogenised) former panels 4, 10 and 13 (old numbering) to include the same number of days. At the same time, we also paid attention at introducing into the sequence a "clear" day in order to better show the effect of the advections. We thus opted to show one week of measurements in each of these figures, as a compromise between completeness and detail (e.g., of the wind velocity field). The new plots extend from 25 to 31 August 2015 (case 1), from 24 to 30 January 2017 (case 2) and from 24 to 30 May 2017 (case 3), respectively. This allows to better appreciate the difference between event- (clear) and non-event (polluted) days.

Moreover, for each episode, the information on the respective seasonal average concentrations were added to the text to provide reference values. Also note that a rigorous assessment of the long-term impact of the phenomenon presented in this part 1 is indeed the purpose of our companion paper (Diémoz et al., 2019) based on a statistical analysis of the complete dataset (2015–2017).

**RC2. FARM leads to too low PM10 concentration in summer and in spring, but seems to work better (even if it does not reproduce the diurnal cycle correctly) in winter (case study 2). Is there an explanation for this difference between summer and winter?**

AR2. We recognise that the results of case study 2 could give the mistaken impression that FARM works better in winter. However, this episode is not representative of a more general behaviour, as thoroughly explained in our companion paper. The latter specifically address the long-term evaluation of the phenomenon described in this "part 1" of the study. We anticipate here Fig. 17 of the companion paper, showing that the discrepancies between simulated and observed $PM_{10}$ concentrations at the surface are larger in winter compared to the other seasons. This is to ascribe to an underestimation of the $PM_{10}$ emission sources from outside the boundaries of the domain (in the second row, the comparison looks better if this external contribution is multiplied by a factor $W=4$).

[Figure]

**Figure 17 (companion paper):** Differences between simulated and observed $PM_{10}$ concentrations at the surface. The mean bias error (MBE) for each case is reported in the plot titles. First row: FARM simulations as currently performed in ARPA for the Donnas (a) and Aosta–Downtown (b) stations. Second row: the $PM_{10}$ concentrations from outside the boundaries of the domain were multiplied by a factor $W=4$.

Section 4.4 of the revised manuscript was slightly modified to anticipate this issue: *The emission inventory used within FARM likely underestimates the real emissions, as also reported in other cases, and for different models, in the scientific literature (e.g., EMEP, 2016; Uchino et al., 2017). In particular, the boundary conditions could not be accurate enough for our aims owing to the abrupt change of the national emission inventory grid resolution (12 km) to the local scale (1 km). This issue can affect the comparison between the model (dry $PM_{10}$) and surface measurements, especially in winter, as discussed more extensively in the companion paper (Diémoz et al., 2019).*

**RC3. Why FARM regularly has time shift in its estimation (for example in Fig. 13 for case study 3)? Fig. S10 shows that modification of the PM concentration from the boundary conditions and the modification of the hygroscopic growth have no impact on the time of the aerosol increase.**

AR3. The reviewer raised an important issue, also mentioned by referee #4 (RC11). We therefore further investigated this aspect, added a figure (S13 in the Supplementary) and updated Sect. 4.4 to include a discussion of this phenomenon:

*As already mentioned in the description of the three cases, the model: a) underestimates the PM mass both in the layer aloft and at the surface, and b) anticipates the peak concentrations compared to the profiles from the ALC. A variety of (possibly concurrent) reasons can explain the observed underestimation [...] The previous considerations, however, fail at comprehensively explaining the time shifts sometimes noticeable between the model and the measurements, i.e. anticipation of the advection arrival time (even in "dry" conditions in the afternoon on the first day of each sequence) and of the layer disappearance in the morning (where hygroscopicity may have an important role). Although an accurate assessment would require a more sophisticated set of instruments to characterise the vertical profile of the wind velocity, here we formulate some hypotheses:*

1. *the NWP model likely anticipates and overestimates the easterly thermally-driven winds in the first hours of the afternoon. This is noticeable, for example, in Fig. S13, where the zonal component of the wind from both COSMO and the surface measurements for case study 1 (August 2015) is plotted, and, on a longer statistical basis, in Fig. S1(b,c), showing that the model has the tendency to see easterly winds more often and with higher intensity compared to the observations. A possible reason for that is the smoothed valley orography used in the NWP model compared to the real one. This is displayed in Fig. S14, showing the difference of the Digital Elevation Model (DEM) used within COSMO and a more realistic DEM (10 m resolution): both valleys and mountain crests are clearly smoothed out by COSMO, with absolute differences well $> 500$ m (and up to 1000 m). This difference could additionally explain why the altitude of the entrainment zone (i.e., the boundary between the free atmosphere and the boundary layer where the thermally-driven circulation develops) is underestimated by COSMO compared to the height of the aerosol layer detected by the ALC;*

2. *COSMO overestimates the nighttime drainage winds (katabatic winds), as noticeable, again, from Fig. S13 for case study 1. This might trigger enhanced cleansing of the lower atmospheric layers during the night as simulated by FARM (see, e.g., the supplementary video file, https://doi.org/10.5446/38391), but undetected by the ALC. An overestimation of the drainage winds would also explain the differences between the simulated and measured daily cycle of specific humidity [...]*

[Figure]

**Figure S13:** Zonal component of wind velocity during episode 1 (August 2015) from COSMO (1000 and 2000 m a.s.l.) and two surface stations (Aosta–Saint Christophe and Saint-Denis). Positive *U* represent wind from the west, negative *U* wind from the east.

[Figure]

**Figure S14:** Difference between the (smoothed) Digital Elevation Model (DEM) used by COSMO-I2 (2.8 km resolution) and a higher-resolution DEM ("real topography", 10 m resolution).

Conclusions were also updated to underline this issue:

*Our investigation allowed an evaluation of the FARM model. Notably, FARM could reproduce the observed arrival of elevated aerosol layers and it correctly attributed them to sources external to the Aosta Valley. However, absolute values of PM concentrations and the timing of the advections were poorly reproduced, with underestimations of aerosol concentrations and time anticipations compared to the measurements. On the basis of a sensitivity study, the former issue may be partly attributed to both water uptake by highly-hygroscopic particles, not fully taken into account in the model, and deficiencies in the emission inventories, especially owing to the coarse resolution of the national one (12 km). As for the timing discrepancies, suboptimal performances of the NWP model to simulate daytime (thermally-driven) and*

*nighttime (katabatic) winds are the most likely sources of error. Despite these limitations, FARM brought insights on the phenomenology addressed, supporting the observations and helping to interpret them. On the other hand, the observation-based results of this work could drive the improvement of the emission inventories, thus enhancing the reliability of the CTM (e.g., Diémoz et al., 2019). In turn, this could allow extending the findings of this work to a wider domain, not covered (or not fully covered) by observations.*

**RC4. P. 1 line 14: "and hygroscopic": the begin of the sentence is a comparison, so that the meaning of this last 2 words is not obvious.**

AR4. Thank you for pointing this out. The sentences now reads: *Results also indicate that the aerosol advected from the Po Valley is hygroscopic, smaller in size and less light-absorbing compared to the aerosol type locally-emitted in the northwestern Italian Alps.*

**RC5. P. 1–2, lines 22–1: for clarity purpose I would add a TO: "likely owing to deficiencies in the emission inventory and TO particle water uptake not fully taken into account"**

AR5. Done.

**RC6. Figure 4d + p. 19 lines 2–7: 1) "PM concentration from non-local sources is represented by the coloured background and the effect of local sources by the contour line, at logarithmic steps;" The logarithmic steps are hardly visible on Fig. 4d (I had to discover that they exist on Fig. 10 to find them on Fig. 4). Could you perhaps use a color scheme with a legend? Without this, it is not evident that the non-local sources exceed the local ones.**

AR6. To address this problem highlighted by the referee, we modified the figures in the following way: a) we made the lines thicker; b) we used different line styles (rather than an additional colour legend) and reported the corresponding concentrations in a new legend. An example is provided below, in Fig. 9 (panel d).

[Figure]

**Figure 9 (panel d):** Case study of 24–30 January 2017. (d) Mass concentration ($PM_{10w}$) from FARM. PM concentration from non-local sources is represented by the coloured background and the effect of local sources by the contour line, at logarithmic steps (dotted: 0.1 $\mu gm^{-3}$; dashed: 1 $\mu gm^{-3}$; continuous, near the surface: 10 $\mu gm^{-3}$).

**RC7. P. 16 line 11: delete the "," after "scattering ratio" + give some indication on the altitude of the described layer.**

AR7. Done, the sentence was modified as follows: *The appearance of the PBL layer is clearly noticeable on 26 August as an increase in the backscatter coefficient up to an altitude of 3 km a.s.l., with scattering ratios SR≃ 4 at midday (light-blue area in the figure) almost doubling (SR> 8, orange-yellow) in a few hours.*

**RC8. P. 16 lines 12-19: I do not really understand your description: from the end of 26th of September, the lowest layer measured by ALC already shows a SR of about 6, so that the "lower levels, with potential consequences on the surface air quality" are already impacted before sunrise on the 27th of September. Your description does not really describes why the SR decrease in the middle of the days. Is the aerosol rich layer entering the ML and contributes then to higher aerosol load in this ML (not observed in Fig. 4a), is it dispersed to higher altitude due to the thermal convection or dispersed horizontally? Please clarify.**

AR8. We believe that the reviewer is referring here to the case of August 26-27 (not September). Indeed, we clearly stated in the text that *a thick aerosol layer is detected by the ALC over the Aosta–Saint Christophe observatory from the afternoon of 26 August,* thus well before the sunrise of August 27, and further specified that *the appearance of the PBL layer is clearly noticeable on 26 August as an increase in the backscatter coefficient up to an altitude of 3 km a.s.l., with scattering ratios SR≃ 4 at midday (light-blue area in the figure) almost doubling (SR> 8, orange-yellow) in a few hours.*

With respect to the missing explanation of the SR decrease in the middle of the day, this was because the interpretation of this ALC-observed behaviour was left to the following paragraphs. However, following the referee's comment, we modified the relevant sentence as follows:

*On 27 August, the ALC backscatter is then observed to decrease in the central part of the day and to increase again in the afternoon. This behaviour keeps very regular for almost a week, with the aerosol-rich layer extending from ground up to 3–3.5 km. As further discussed in the next paragraphs, we anticipate here that the main factor driving this cycle is likely an enhanced hygroscopic growth of aerosol advected from the Po Valley from the afternoon to the early morning, this effect also leading to formation of low clouds within the aerosol layer at night (screened out as white areas in the figure). In fact, the transition from aerosol to the cloud phase is very sharp, as also noticeable from the sudden increase of more than 40 $Wm^{-2}$ of the downward infrared irradiance monitored at the same site.*

**RC9. P. 19: is it possible that the observed shift (from some hours to 1/4 of a day) of the maximum concentration between FARM and ALC can be due to an overestimation of the local effects in comparison to the non-local ones?**

AR9. Local effects can indeed impact on the daily evolution of the concentrations. However, we believe that the shift observed in this case is mainly triggered by suboptimal simulations of the wind fields by the NWP model, as explained at point AR3 of the present document.

**RC10. Figure 8 and p. 6 lines 18–20 + p. 20 lines 13–15: the measured PM2.5 and PM10 during the case study are similar to the annual mean concentration (p.6). An increase of PM2.5 and PM10 is clearly visible from the 26th of August to the beginning of September (at least at Aosta), but the readers cannot be sure that this does not correspond to a usual fluctuations of the aerosol load. Similarly, it is not really visible that FARM predicts an increase of the PM10 concentration at Aosta during the case study.**

AR10. Following this comment, we modified the relevant sentence as follows: *An increase in daily concentrations (up to maximum values of 10 $\mu g m^{-3}$ for $PM_{2.5}$ and 16–22 $\mu g m^{-3}$ for $PM_{10}$) can be clearly noticed at both sites, leading to concentrations slightly higher than average for the same period (7 $\mu g m^{-3}$ for $PM_{2.5}$ in Aosta–Downtown and 12 $\mu g m^{-3}$ for $PM_{10}$ at both sites, considering the 2015–2017 series). The statistical significance of such PM increases during these transport episodes, compared to the natural variability of the aerosol load in non-advection conditions, is assessed in the companion paper using the full long-term dataset.*

Moreover, as already mentioned in AR1, the seasonal average concentrations were added to the text to provide reference values.

**RC11. Fig. 4e + page 20 lines 8–9: the PM10 daily cycle does not correspond to the Po Valley air masses transport seen by the ALC. The given explanation relates to the mass losses occurring in TEOM due to secondary aerosol volatility. Is it expected that this volatility should be different between nighttime and daytime (i.e. larger during the night) in order to explain the different diurnal cycles?**

AR11. This is an important remark. We therefore decided to update the manuscript to discuss this issue in further detail.

Section 3.1.5: *Two Tapered Element Oscillating Microbalance (TEOM) 1400a monitors (Patashnick and Rupprecht, 1991) are used for continuous measurements of $PM_{10}$ hourly concentrations at the stations of Aosta–Downtown and La Thuile. These instruments are not compensated for mass loss of semi-volatile compounds (Green et al., 2009) and could be insensitive to specific compounds, such as ammonium nitrate (e.g., Charron et al., 2004), which leads to underestimations, especially in the cold season, compared to the SM200. Conversely, overestimations by the TEOM compared to daily averages from the SM200 reference instrument are found in summer and are not fully understood at present. Therefore, TEOM monitors are only employed here for qualitative estimates of short-term variations of the aerosol burden while daily-averaged concentrations will be only taken from the SM200 instruments.*

Section 4.1.3: *To evaluate the impacts on surface air quality parameters during the episode, hourly $PM_{10}$ concentrations at the surface as measured in Aosta–Downtown and simulated by FARM in Aosta–Saint Christophe are presented in Fig.3e ($PM_{10}$ monitoring at La Thuile was not yet operational, at that time). Apart two spikes (80 $\mu g m^{-3}$) on 25 and 26 August (presumably of local origin), the concentrations measured in Aosta–Downtown by the TEOM show a slight increase after the arrival of the layer, but without sudden jumps. Also, PM concentrations are generally higher during daytime compared to the night, according to the expected cycle of the summertime local sources (e.g., traffic, resuspension, etc.). This features, however, can be connected to the fact that mass loss occurs in TEOM due to secondary aerosol volatility, as better discussed in the companion paper by comparing the daily $PM_{10}$ cycle from this instrument and the Fidas OPC in Aosta–Saint Christophe. Moreover, this volatility could be different between nighttime and daytime, which would also contribute to the observed daily behaviour. Besides, FARM estimates at the surface are again lower than measurements (-60%, on average). Daily $PM_{10}$ concentrations observed by Opsis SM200 instruments during the case study in Aosta–Downtown and Donnas are shown in Fig. 7, which includes the whole episode (correlation index with TEOM measurements $\rho = 0.84$).*

**RC12. Fig. S5 is cited before Fig. S4**

AR12. Corrected, thank you.

**RC13. P. 25 line 2: it has however to be mentioned (and perhaps explained) that the wind direction at Saint-Denis (Fig. S4e), after turning to the east during the afternoon of the 26th of January, turns again to west on the 27th in the morning and stay globally at west during the rest of the case study.**

AR13. To better show the different flow regimes during this episode, we made some integrations to the analysis. First, wind direction was plotted in Fig. S4 only for cases when wind speeds exceeds $1\,\mathrm{m\,s}^{-1}$ to exclude calm wind conditions. Secondly, we further considered the wind measurements in Donnas, where the wind regime change is much clearer, owing to the weaker temperature inversion. Table 1 (used instruments) was modified accordingly.

[Figure]

**Figure S4:** Case study of 26–29 January 2017. (a) Same as in Fig. 9 in the main paper; (b) Surface relative humidity, specific humidity and temperature measured at the Aosta–Saint Christophe weather station; (c) Particle number distribution from the Palas optical counter; (d) Particle number concentration (sum of all channels) from the Palas optical counter; (e) Wind speed and direction at the Saint-Denis station (800 m a.s.l); (f) Wind speed and direction at the Donnas station (316 m a.s.l). No data from the photometer are available for the selected period, since the instrument was not operating.

Given these changes, Sect. 4.2.2 now reads:

*The wind field over Aosta–Saint Christophe, depicted in Fig. 9a, presents a very different pattern compared to the first case addressed (Fig. 3a). Firstly, calm wind is measured for the whole period at the bottom of the valley. This is due to a shallow temperature inversion in the lower atmospheric layers in the main valley. Conversely, at the Saint-Denis station, located above the inversion layer, and at the Donnas station, where the temperature inversion is weaker, the wind pattern is more representative of the wider circulation: for example, the average wind speed in Saint-Denis is about 4 $ms^{-1}$ on 26 January afternoon (Fig. S4e) and*

*the wind clearly turns from west (morning) to east (afternoon), simultaneously with the appearance of the layer. The same wind change is detected in Donnas on the same day (Fig. S4f), with easterly wind speeds $> 1\ ms^{-1}$ for several hours in the afternoon. As a further difference with the first case, the forecasted wind at 1000–2000 m a.s.l. does not show any change in direction typical of the thermal winds. For example, at 2000 m a.s.l. the circulation is continuous, and vigorous (up to 6 $ms^{-1}$), from the afternoon of 26 to the beginning of 29 January. Indeed, this winter case study interestingly shows that thermally-driven winds are not the only mechanism, especially in winter, driving the advection of air masses from the Po Valley to the Alps. Rather, the synoptical circulation can push the air masses towards the Alpine valleys, as in this case. In fact, the flow clearly reveals its southern origin at elevations above the mountain crest (e.g., 3000 m a.s.l.), where the wind is not channelled within the main valley. At that altitude, the wind speed is even greater than 20 $ms^{-1}$. Finally, on 29 January, the measurements in Saint-Denis (gradual increase of the speed of westerly wind) and in Donnas (even stronger wind, again from the west), and COSMO simulations (wind reversal at 1000–2000 m a.s.l) correlate with the disappearance of the layer better than observations performed at the bottom of the valley (calm wind). [...]*

**RC14. P. 25 lines 10–12: the increase of the wind speed at Saint-Denis on the 29th is not very large (about 2 m/s on the 26th in the morning, increase to 4 m/s on the 26th in the afternoon, about 1 m/s until the end of the 28th, then 2 m/s), so that the correlation with the disappearance of the layer is not really explicit.**

AR14. Please, refer to AR13.

**RC15. Fig. S6 + p. 25 lines 13–15: the described transition (for the lowest levels I suppose) is quite difficult to see: too much backtrajectories, the lowest ones being represented under the highest ones. A figure allowing to see not only the trajectories but also the change of the altitude over the Po Valley (lines 15–29) would perhaps be more interesting).**

AR15. Following this comment, and the ones by referee #1 (RC3) and #4 (RC7), we modified the back-trajectory figures in the following way: a) we plotted in separate panels trajectories ending at altitudes < 2000 and > 2500 m a.s.l. over Aosta–Saint Christophe; b) to further simplify the figures, we only show back-trajectories for specific times corresponding to the most significant variations of circulation patterns; c) for each time selected we added a bottom panel with the trajectory altitude along their journey. An example of the new plots now included in the paper is provided in Fig. 5.

[Figure]

**Figure 5:** 48-hours back-trajectories ending at Aosta–Saint Christophe on 26 August 2015 at 18 UTC at altitudes lower than 2000 m a.s.l. (a) and higher than 2500 m a.s.l. (b). The trajectories are cut at the border of the COSMO model. The colour scale represents the back-trajectory arrival height. Corresponding altitudes of the backtrajectories vs time are reported in the bottom panels. The dots along each trajectory mark a 1-hour step and the black star indicates the trajectory arrival point (Aosta–Saint Christophe).

**RC16. Fig. S4d: the particle number concentration increase during the case study is clearly visible. It is however not possible to determine if this is a peculiar or a normal event occurring regularly during winter.**

AR16. As explained in AR1, the duration of the phenomenon represented in the figure was extended to one week in order to include more "non-event" days and better show the effect of the advections on the "standard" conditions in Aosta. The text in Sect. 4.2.3 was also updated in the following way: *The instrument reveals a notable increase in the number concentration for particles smaller than 0.5 μm (Fig. S4c) in coincidence to the arrival of the aerosol layer. The total number concentration (Fig. S4d) gradually increases from few hundreds particles $cm^{-3}$ up to 3000 particles $cm^{-3}$ and decreases again on 29 January (the average value in winter 2016–2017 in conditions of local pollution being 650 particles $cm^{-3}$).*

**RC17. Fig. S9 is cited before fig. S8.**

AR17. Corrected.

**RC18. P. 32–33, lines 31–6:**The hygroscopic growth of the aerosol can clearly be a cause of the discrepancy between FARM and ALC, even if FARM is also not able to reproduce PM measurements and their diurnal cycles. The difference of the measured size fraction (TSP for ALC and PM10 for FARM does however not seem to be really relevant since: 1) Aerosol from the Po Valley are described as small ones in the paper, see for ex. Case study 2, where the increase in PM10 is similar to the one of PM2.5 (Fig. 12), 2) the size distribution measured by the OPC and sun photometer peak in the accumulation mode and do not show big particles.

AR18. The referee is right, we modified the relevant sentence as follows:

*A variety of (possibly concurrent) reasons can explain the observed underestimation, mainly related to:*

1. *inaccuracies in retrieving the PM concentration from the ALC backscatter. As mentioned, ALC measures aerosol backscatter, so that specific tools were developed (Dionisi et al., 2018) and are used here to associate a PM value to it. Still, the expected error associated to these estimates is of the order of 30–40%. In addition, the ALC retrieval is based on functional relationships derived assuming maximum RH of 95%. Higher FARM-ALC discrepancies can be expected when RH>DRH, and particularly at RH>95%;*

2. *inaccuracies in the CTM simulations [...]*

**RC19. P. 33 lines 7–13: How good is the estimation of the PBL height in the CTM simulations? Could a bad estimation of the PBL height be also a source of inaccuracies?**

AR19. This could be a further, interesting evaluation of the model output to be performed by comparing the mixing layer height (Hmix) of the model pre-processor (SURFPRO) to the ALC measurements. In fact, these instruments are generally quite powerful in showing the temporal evolution of the mixing layer using the aerosol as tracer (e.g., Angelini et al., 2009). According to this comment/suggestion we modified Sect. 4.4 as follows:

*Finally, some underestimation of PM values could also be due to overestimation of the FARM simulated PBL height. An evaluation of this kind of effect is in principle possible by comparing the simulations with the ALC-derived PBL height (e.g., Angelini et al., 2009; Haeffelin et al., 2012). However, this should be performed only selecting non-advection conditions, i.e., those in which the ceilometer signal is only affected by local aerosols and is thus able to follow the daily evolution of the local PBL using local particles as tracers. This kind of investigation was, however, beyond the scope of the present work. Challenges and recent efforts to define a PBL in mountainous areas ("Mountain Boundary Layer", MBL) and discrepancies between the MBL and the aerosol layer are more extensively described by Lehner and Rotach (2018).*

**RC20. P. 34 line 30–32: as explained previously, I do not see any evidence of a residual layer sinking towards the surface in this study.**

AR20. Following the referee's comment, all references about sinking residual layers were removed from the text.

**Response to Anonymous Referee #4**

**General comments: The paper analyzes the impact of pollutants transport from the Po Valley to the air quality on the Alpine region. Three selected cases studies of transport over the Aosta Valley are selected among a 3 years period and analyzed in detail by means of a wide- ranging set of observations and numerical simulations. Moreover, the observational data are used to evaluate the performances of the FARM chemical transport model. These results provide an interesting and comprehensive description of the complex phenomena of mountain-valley exchange and deserve therefore to be published.**

We thank the reviewer for taking the time to revise our manuscript and for his/her pertinent comments. Our reply to these is given hereafter (the text in italics represents a citation of the revised manuscript and the figure references follow the updated numbering).

**Referee's comment 1. Even if the paper is well written, though, the large amount of data presented in both the main paper and the support material makes the manuscript dispersive and fragmented. I would suggest to further select the figures that are useful to support their message and eliminate redundancies. For example, I invite the authors to carefully evaluate if the section 4.1.5 is bringing any relevant information: the analysis seems not fully convincing, is limited to a single case and the information that intends to bring is already conveyed by the FARM model. Similarly, some figures can be just substituted by a short mention in the text (I would reconsider the relevance of fig. 3, S2c, S4e, S5, S12).**

Author's response 1. We understand the referee's objection. At the same time, we believe that the evaluation of the spatial extent of the phenomenon under investigation is important for its understanding. In this respect, Sect. 4.1.5 is the only one in which the phenomenon, which is carefully evaluated locally (Aosta region) with a large set of in situ and ground based measurements, is **observed** (not simulated) over the wider Northern Italy domain using space-based data. We therefore believe that this section, although with some limitations, is an important **observation-based** support to the FARM **model** results. For this reason, we would prefer to keep it.

We are also aware that the manuscript has a high number of figures to show results and support the relevant discussion, including several points raised by the reviewers. That's why we made the choice to leave additional, supporting material out of the main text (Supplement). We believe this choice still allows a straightforward reading of the main text, while providing additional details to those readers interested in more specific aspects of the study.

Still, following the reviewer's advice, we removed Fig. 3 from the text as most information was already

available in Fig. 4a.

**RC2. Page 2, line 19–22: Basing on the results of the FARM comparison I would rather highlight that the paper aimed at an evaluation of the model and then mentioning that, despite its limitations, it still brought insights on the cases, supporting the observations. Similarly, I would rephrase the paragraph in the conclusions, accordingly (Page 35, line 14 onward)**

AR2. Both the abstract and the conclusions were rephrased following the reviewer's advice.

Abstract: *Results show that the simulations are important to the understanding of the phenomenon under investigation. However, in quantitative terms, modelled $PM_{10}$ concentrations are 4–5 times lower than the ones retrieved from the ALC and maxima are anticipated in time by 6–7 hours. Underestimated concentrations are likely mainly due to deficiencies in the emission inventory and to water uptake of the advected particle not fully reproduced by FARM, while timing mismatches are likely an effect of suboptimal simulation of up-valley and down-valley winds by COSMO.*

Conclusions: *Our investigation allowed an evaluation of the FARM model. Notably, FARM could reproduce the observed arrival of elevated aerosol layers and it correctly attributed them to sources external to the Aosta Valley. However, absolute values of PM concentrations and the timing of the advections were poorly reproduced, with underestimations of aerosol concentrations and time anticipations compared to the measurements. On the basis of a sensitivity study, the former issue may be partly attributed to both water uptake by highly-hygroscopic particles, not fully taken into account in the model, and deficiencies in the emission inventories, especially owing to the coarse resolution of the national one (12 km). As for the timing discrepancies, suboptimal performances of the NWP model to simulate daytime (thermally-driven) and nighttime (katabatic) winds are the most likely sources of error. Despite these limitations, FARM brought insights on the phenomenology addressed, supporting the observations and helping to interpret them. On the other hand, the observation-based results of this work could drive the improvement of the emission inventories, thus enhancing the reliability of the CTM (e.g., Diémoz et al., 2019). In turn, this could allow extending the findings of this work to a wider domain, not covered (or not fully covered) by observations.*

**RC3. Page 4, line 3: Do you really mean that is the partition between local and non-local sources that help to assess the impact of air pollutants on health, climate and ecosystem?**

AR3. No, we meant to say that in order to set up mitigation actions to limit the impact of air pollutants on health, climate and ecosystems it is first necessary to understand where these are originated. Thank you for pointing out that the sentence was misleading, we rephrased it as follows:

*Over the impacted areas, a correct partitioning between local and non-local sources is therefore necessary to 1) correctly interpret the exceedances of air quality limits; 2) develop joint efforts and large-scale mitigation strategies (WMO, 2012) to reduce the frequency and impact of pollution episodes on citizen health (Straif et al., 2013; WHO, 2016; Zhang et al., 2017), climate (Clerici and Mélin, 2008; Lau et al., 2010; Zeng et al., 2015) and ecosystems (Carslaw et al., 2010; Bourgeois et al., 2018; Burkhardt et al., 2018).*

**RC4. Page 14, line 21. "Therefore, to reduce errors in trajectories with increasing running time, we limit the computation to this duration". The sentence between commas, in this context, is misleading. Rewrite as "Therefore, we limit the computation to this duration"**

AR4. Done.

**RC5. Page 18, Figure 6. It would be more meaningful to show the trajectories at 18:00 UTC for this case, since at this time you observe the start of the intense layer arrival and the trajectories are also showing a larger impact from the Po Valley.**

AR5. The figure (Fig. 5, with the new numbering) was modified according to the referee's comment. It is shown here below (cf. also AR7 for further updates to the figure):

[Figure]

**Figure 5:** 48-hours back-trajectories ending at Aosta–Saint Christophe on 26 August 2015 at 18 UTC at altitudes lower than 2000 m a.s.l. (a) and higher than 2500 m a.s.l. (b). The trajectories are cut at the border of the COSMO model. The colour scale represents the back-trajectory arrival height. Corresponding altitudes of the backtrajectories vs time are reported in the bottom panels. The dots along each trajectory mark a 1-hour step and the black star indicates the trajectory arrival point (Aosta–Saint Christophe).

**RC6. Page 16, line 14: Mention here that you see these surface impacts in your measurements (Fig. 8). Also, what will happen if you average the hourly measurements of TEOM on a daily interval? Will it be consistent with the results of figure 8? Please note that the time interval shown in figure 4 seems to be, compared to figure 8, in the middle of the event (Starting from 26 August and ending at the end of 3 September), so that is not possible to identify any difference with the previous or following phase.**

AR6. The discussion about the TEOM has been refined in the updated manuscript, also following the

comment by referee #3 (RC11).

Section 3.1.5: *Two Tapered Element Oscillating Microbalance (TEOM) 1400a monitors (Patashnick and Rupprecht, 1991) are used for continuous measurements of $PM_{10}$ hourly concentrations at the stations of Aosta–Downtown and La Thuile. These instruments are not compensated for mass loss of semi-volatile compounds (Green et al., 2009) and could be insensitive to specific compounds, such as ammonium nitrate (e.g., Charron et al., 2004), which leads to underestimations, especially in the cold season, compared to the SM200. Conversely, overestimations by the TEOM compared to daily averages from the SM200 reference instrument are found in summer and are not fully understood at present. Therefore, TEOM monitors are only employed here for qualitative estimates of short-term variations of the aerosol burden while daily-averaged concentrations will be only taken from the SM200 instruments.*

Section 4.1.3: *To evaluate the impacts on surface air quality parameters during the episode, hourly $PM_{10}$ concentrations at the surface as measured in Aosta–Downtown and simulated by FARM in Aosta–Saint Christophe are presented in Fig.3e ($PM_{10}$ monitoring at La Thuile was not yet operational, at that time). Apart two spikes (80 $\mu g\,m^{-3}$) on 25 and 26 August (presumably of local origin), the concentrations measured in Aosta–Downtown by the TEOM show a slight increase after the arrival of the layer, but without sudden jumps. Also, PM concentrations are generally higher during daytime compared to the night, according to the expected cycle of the summertime local sources (e.g., traffic, resuspension, etc.). This features, however, can be connected to the fact that mass loss occurs in TEOM due to secondary aerosol volatility, as better discussed in the companion paper by comparing the daily $PM_{10}$ cycle from this instrument and the Fidas OPC in Aosta–Saint Christophe. Moreover, this volatility could be different between nighttime and daytime, which would also contribute to the observed daily behaviour. Besides, FARM estimates at the surface are again lower than measurements (-60%, on average). Daily $PM_{10}$ concentrations observed by Opsis SM200 instruments during the case study in Aosta–Downtown and Donnas are shown in Fig. 7, which includes the whole episode (correlation index with TEOM measurements $\rho$ = 0.84).*

Regarding the second comment by the referee about the time interval shown in former Fig. 4, we followed the reviewer's suggestion and similar remarks from referee #2 (RC2) and referee #3 (RC1 and RC10). Thus, we expanded (and homogenised) former panels 4, 10 and 13 (old numbering) to include the same number of days. At the same time, we also paid attention at introducing into the sequence a "clear" day in order to better show the effect of the advections. We thus opted to show one week of measurements in each of these figures, as a compromise between completeness and detail (e.g., of the wind velocity field). The new plots extend from 25 to 31 August 2015 (case 1), from 24 to 30 January 2017 (case 2) and from 24 to 30 May 2017 (case 3), respectively. This allows to better appreciate the difference between event- (clear) and non-event (polluted) days.

Moreover, for each episode, the information on the respective seasonal average concentrations were added to the text to provide reference values. Also note that a rigorous assessment of the long-term impact of the phenomenon presented in this part 1 is indeed the purpose of our companion paper (Diémoz et al., 2019) based on a statistical analysis of the complete dataset (2015–2017).

**RC7. Page 17, line 33: Do you mean here that the transport from the Po Valley persists at every hour along the following days? Indeed, considering the evolution of the layer as seen from the measurements, it would be useful to show which is the typical transport during the whole event at different hours, for example adding the average position of the trajectories (or the PDF of the position in case of large variability) on the considered period. If the authors want to still keep the trajectory of the first day of the episode as a reference for the layer arrival, I would suggest, to avoid burden the paper with additional plots, showing just the synoptic hours instead of a 3-hour time step; it should still be enough to demonstrate the transport evolution. This would also allow comparing the first and third transport type with the mean transport behaviour of the second case study, when the layer persists for the whole duration of the event.**

AR7. The sentence at line 33 was modified as follows: *The analysis of the corresponding back-trajectories confirms that transport of polluted air masses from the Po basin also occurs in the afternoons of the other days of this episode, until the flux changes again to a north-western configuration (Fig. S3c and f).*

Following the comment by the reviewer (and similar remarks from referees #1 (RC3) and #3 (RC15)), the back-trajectories figures were modified in this way: we plotted in separate panels trajectories ending at altitudes < 2000 and > 2500 m a.s.l. over Aosta–Saint Christophe. Also, to further simplify the figures, we only show back-trajectories for specific times corresponding to the most significant variations of circulation patterns. Finally, according to referee #3 (RC15), for each time selected we added a bottom panel with the trajectory altitude along their journey. An example of the new plots is provided in Fig. 5, see AR5.

**RC8. Page 21, line 15: Can you add some reference for the typical Angstrom exponent for the Aosta valley in comparison?**

AR8. Done, we added the following sentence: *These values should be compared to the lower Ångström exponents typically measured in the Aosta Valley, i.e. ~1.1 on average (Diémoz et al., 2014, 2019).*

**RC9. Page 25, line 18 (but same for page 30, line 6): How do you estimate this residence time, is it an average of the time spent by each trajectory over the Po Valley at all levels? From the shown figure (even if it is difficult to distinguish), the transport, including the part outside the PBL, seems faster (around 20 hours or even less). Can you be more specific?**

AR9. This was simply computed considering that each dots along each trajectory mark a 1-hour step (as stated in caption of Fig. 5). Given this comment we clarified this point in the text as follows: *The trajectory altitude tends to decrease in the afternoon reaching the elevations of the polluted boundary layer (Fig. S6b), thus leading to effective aerosol transport to the Aosta Valley. In fact, considering that each dot in Fig. S6b represents a 1-hour step, we estimate a mean air masses residence time in the Po Valley PBL of 30–35 hours before arriving over the observing site. Finally, trajectories turn westerly on 29 January, in agreement with the removal of the layer over Aosta (Fig. S6c and f).*

**RC10. Page 25, line 33–35: How do you justify these affirmations?**

AR10. The sentence was partially rephrased: *Taking into consideration these different daily evolution patterns and the sources included in the emission inventory, the most likely reasons for the differences between the model and the measurements at the surface appear to be an underestimation of the residential heating (actually switched on all day during these very cold days) and an overestimation of the traffic road contribution, together with an overestimation of the mixing layer height growth at midday by the NWP model.*

**RC11. Page 32, line 29: It would be interesting to see this comparison indeed since, from figure S1, it seems that the model has the tendency to see easterly winds more often and with higher intensity**

**respect to the observations. Is this comparison limited just to the surface measurements? May it be that there are problems in the higher layers (where most of the pollution layer transport is coming from)? For example, in the case study 2, when the winds were weaker both at the ground and at higher layers, the time evolution of the FARM simulation of Figure 10d was in better agreement with the observations. Is also interesting to note that the time evolution of the vertical distribution of PM10 from FARM is in better agreement (especially the local contribution) with the surface TEOM measurements rather than with the elevated layers observations.**

AR11. The reviewer raised some important issues, also mentioned by referee #3 (RC3). We therefore decided to better investigate the capability of COSMO in reproducing the measured wind fields (unfortunately, our instrumentation allows validation only at the surface) and we updated Sect. 4.4 accordingly:

*As already mentioned in the description of the three cases, the model [...] anticipates the peak concentrations compared to the profiles from the ALC [...], i.e. [it shows] anticipation of the advection arrival time (even in "dry" conditions in the afternoon on the first day of each sequence) and of the layer disappearance in the morning (where hygroscopicity may have an important role). Although an accurate assessment would require a more sophisticated set of instruments to characterise the vertical profile of the wind velocity, here we formulate some hypotheses:*

1. *the NWP model likely anticipates and overestimates the easterly thermally-driven winds in the first hours of the afternoon. This is noticeable, for example, in Fig. S13, where the zonal component of the wind from both COSMO and the surface measurements for case study 1 (August 2015) is plotted, and, on a longer statistical basis, in Fig. S1(b,c), showing that the model has the tendency to see easterly winds more often and with higher intensity compared to the observations. A possible reason for that is the smoothed valley orography used in the NWP model compared to the real one. This is displayed in Fig. S14, showing the difference of the Digital Elevation Model (DEM) used within COSMO and a more realistic DEM (10 m resolution): both valleys and mountain crests are clearly smoothed out by COSMO, with absolute differences well > 500 m (and up to 1000 m). This difference could additionally explain why the altitude of the entrainment zone (i.e., the boundary between the free atmosphere and the boundary layer where the thermally-driven circulation develops) is underestimated by COSMO compared to the height of the aerosol layer detected by the ALC;*

2. *COSMO overestimates the nighttime drainage winds (katabatic winds), as noticeable, again, from Fig. S13 for case study 1. This might trigger enhanced cleansing of the lower atmospheric layers during the night as simulated by FARM (see, e.g., the supplementary video file, `https://doi.org/10.5446/38391`), but undetected by the ALC. An overestimation of the drainage winds would also explain the differences between the simulated and measured daily cycle of specific humidity [...]*

[Figure]

**Figure S13:** Zonal component of wind velocity during episode 1 (August 2015) from COSMO (1000 and 2000 m a.s.l.) and two surface stations (Aosta–Saint Christophe and Saint-Denis). Positive $U$ represent wind from the west, negative $U$ wind from the east.

[Figure]

**Figure S14:** Difference between the (smoothed) Digital Elevation Model (DEM) used by COSMO-I2 (2.8 km resolution) and a higher-resolution DEM ("real topography", 10 m resolution).

Conclusions were also updated to underline this issue, as already mentioned in AR2: *As for the timing discrepancies, suboptimal performances of the NWP model to simulate daytime (thermally-driven) and nighttime (katabatic) winds are the most likely sources of error.*

Finally, please note that an in-depth examination of FARM capabilities to reproduce the observed PM concentrations for event and non-event days was done in the companion paper based on the long-term dataset (2015–2017). We anticipate here Fig. 17 (companion paper), showing that the discrepancies between simulated and observed $PM_{10}$ concentrations at the surface are minimum in cases of non-event days, which agrees with the referee's remark.

[Figure]

**Figure 17 (companion paper):** Differences between simulated and observed PM$_{10}$ concentrations at the surface. The mean bias error (MBE) for each case is reported in the plot titles. First row: FARM simulations as currently performed in ARPA for the Donnas (a) and Aosta–Downtown (b) stations. Second row: the PM$_{10}$ concentrations from outside the boundaries of the domain were multiplied by a factor $W$=4.

**RC12. Page 4, line 10: "will be quantified in a companion ..."**

AR12. Done.

**RC13. Page 4, line 28: For easier reading, specify which valley are you talking about. In the same paragraph, you are referring to both the Po valley and the Alpine valleys.**

AR13. Done. The sentence now reads: *[...] the heaviest burden of particles did not come from the largest urban settlement in the Aosta Valley, but rather from outside the region, namely from the Po basin.*

**RC14. Figure 2: The dashed lines are not needed.**

AR14. Removed.

**RC15. Page 9, line 21: "... altitude of the extinction coefficient."**

AR15. Done.

**RC16. Page 21, line 22: Refer also to the figure S8.**

AR16. Reference to the figure was added.

**RC17. Page 34, line 6: Add a ":" after "(see Introduction)"**

AR17. Done.

**RC18. Page 35, line 19: "... these issues may be partly attributed to ..."**

AR18. Done.

**References**

[revised manuscript text omitted]
  at altitudes < 2000 m a.s.l. (a, b and c) and > 2500 m a.s.l. (d, e and f) during the summer episode (August  2015). Before the arrival of the polluted air mass over Aosta all trajectories follow the synoptics circulation from the west (a, d). Later on the same day, the lowest trajectories turn and cross the Po basin, leading to advection of polluted air over Aosta, while the highest ones still come from the west (b, e). At the end of the episode, all trajectories turn back to the west (c, f). The trajectories are cut at the border of the COSMO model. The colour scale represents the arrival height. The dots along each trajectory mark a 1-hour step.

**S5 Supporting material to case study 2**

[Figure]

Figure S4: Case study of 26–29 January 2017. (a) Same as in Fig. 9 in the main paper; (b) Surface relative humidity, specific humidity and temperature measured at the Aosta–Saint Christophe weather station; (c) Particle number distribution from the Palas optical counter; (d) Particle number concentration (sum of all channels) from the Palas optical counter; (e) Wind speed and direction at the Saint-Denis station (800 m a.s.l); (f) Wind speed and direction at the Donnas station (316 m a.s.l). No data from the photometer are available for the selected period, since the instrument was not operating.

[Figure]

Figure S5:  Profile of pseudo-equivalent potential temperature measured along the mountain slope on January 2017. A weakening of the temperature inversion, and a more mixed boundary layer, are clearly detected from 26 to 28 January, i.e. during the advection episode.

[Figure]

Figure S6: 48-hours back-trajectories ending at  altitudes < 2000 m a.s.l. (a, b and c) and > 2500 m a.s.l. (d, e and f) during the winter episode (January 2017): (a, d) morning of the advection day, (b, e) after the arrival of the layer over Aosta, and (c, f) end of the episode. Before the arrival of the layer, the trajectories are higher than the polluted mixing layer over the Po basin (a), whilst their altitude progressively decrease (b), leading to a more effective transport of pollution.

[Figure]

Figure S7: Daily PM$_{10}$ surface (2D) simulations from FARM over the Aosta Valley (background colour) and in-situ measurements (circles) for case sudy 2 (January 2017).

**S6    Supporting material to case study 3**

[Figure]

Figure S8: 48-hours back-trajectories ending at Aosta–Saint Christophe at altitudes < 2000 m a.s.l. (a, b and c) and > 2500 m a.s.l. (d, e and f) during the spring episode (May 2017): (a, d) morning of the advection day, trajectories following the synoptic circulation; (b, e) evening of the same day, lowest trajectories diverted owing to the breeze circulation and crossing the Po basin; (c, f) end of the episode, all trajectories coming from north-west.

48-hours back-trajectories ending at Aosta–Saint Christophe on 25 May 2017.

[revised manuscript text omitted]

**S8  Inaccuracies of wind velocity simulations by the NWP model**

[Figure]

Figure S13: Zonal component of wind velocity during episode 1 (August 2015) from COSMO (1000 and 2000 m a.s.l.) and two surface stations (Aosta–Saint Christophe and Saint-Denis). Positive $U$ represent wind from the west, negative $U$ wind from the east.

[Figure]

Figure S14: Difference between the (smoothed) Digital Elevation Model (DEM) used by COSMO-I2 (2.8 km resolution) and a higher-resolution DEM ("real topography", 10 m resolution).

[Figure]

Figure S15: Comparison among specific humidity measured at Aosta–Saint Christophe and simulated by COSMO at two different altitudes (surface and 2000 m a.s.l.).

**References**

[revised manuscript text omitted]